# Brain high-throughput multi-omics data reveal molecular heterogeneity in Alzheimer's disease

**Abdallah M. Eteleeb**[1,2], **Brenna C. Novotny**[1], **Carolina Soriano Tarraga**[1], **Christopher Sohn**[1], **Eliza Dhungel**[3], **Logan Brase**[1], **Aasritha Nallapu**[1], **Jared Buss**[1], **Fabiana Farias**[1,4], **Kristy Bergmann**[1,4], **Joseph Bradley**[1,4], **Joanne Norton**[1,4], **Jen Gentsch**[1,4], **Fengxian Wang**[1,4], **Albert A. Davis**[5,6], **John C. Morris**[2,5,6], **Celeste M. Karch**[1,2,4,6], **Richard J. Perrin**[2,5,6,7], **Bruno A. Benitez**[8‡], **Oscar Harari**[1,2,6¤‡]*

1 Department of Psychiatry, Washington University, Saint Louis, St. Louis, Missouri, United States of America, 2 The Charles F. and Joanne Knight Alzheimer Disease Research Center, Washington University, St. Louis, Missouri, United States of America, 3 Department of Bioinformatics and Genomics, University of North Carolina at Charlotte, Charlotte, North Carolina, United States of America, 4 NeuroGenomics and Informatics Center, Washington University, St. Louis, Missouri, United States of America, 5 Department of Neurology, Washington University, St. Louis, Missouri, United States of America, 6 Hope Center for Neurological Disorders, Washington University, St. Louis, Missouri, United States of America, 7 Department of Pathology and Immunology, Washington University, St. Louis, Missouri, United States of America, 8 Department of Neurology and Neuroscience, Harvard Medical School and Beth Israel Deaconess Medical Center, Boston, Massachusetts, United States of America

¤ Current address: Department of Neurology, Division of Neurogenetics, The Ohio State University, Columbus, Ohio, United States of America
‡ These authors are joint senior authors on this work.
* oscar.harari@usumc.edu

**Data Availability Statement:** Omics raw data from the Knight ADRC participants are available by request at the NIAGADS Knight ADRC collection

## Abstract

Unbiased data-driven omic approaches are revealing the molecular heterogeneity of Alzheimer disease. Here, we used machine learning approaches to integrate high-throughput transcriptomic, proteomic, metabolomic, and lipidomic profiles with clinical and neuropathological data from multiple human AD cohorts. We discovered 4 unique multimodal molecular profiles, one of them showing signs of poor cognitive function, a faster pace of disease progression, shorter survival with the disease, severe neurodegeneration and astrogliosis, and reduced levels of metabolomic profiles. We found this molecular profile to be present in multiple affected cortical regions associated with higher Braak tau scores and significant dysregulation of synapse-related genes, endocytosis, phagosome, and mTOR signaling pathways altered in AD early and late stages. AD cross-omics data integration with transcriptomic data from an *SNCA* mouse model revealed an overlapping signature. Furthermore, we leveraged single-nuclei RNA-seq data to identify distinct cell-types that most likely mediate molecular profiles. Lastly, we identified that the multimodal clusters uncovered cerebrospinal fluid biomarkers poised to monitor AD progression and possibly cognition. Our cross-omics analyses provide novel critical molecular insights into AD.

(https://www.niagads.org/knight-adrc-collection)
with accession numbers NG00083 (https://www.
niagads.org/datasets/ng00083), NG00102 (https://
www.niagads.org/datasets/ng00102), NG00113
(https://dss.niagads.org/datasets/ng00113/), and
NG00108 (https://dss.niagads.org/datasets/
ng00108/) for transcriptomics (bulk), proteomics,
metabolomics, and single-nuclei RNA-seq
respectively. Access to all individual-level data
requires an approved NIAGADS application.
Documents required for NIAGADS data request can
be found here (https://www.niagads.org/resources/
documents-and-guidelines) and instructions on
how to submit a data access request are available
here (https://www.niagads.org/data/request/data-
request-instructions). MSBB transcriptomics and
proteomics raw data are publicly available at
Synapse under Synapse IDs syn3157743 (https://
www.synapse.org/#!Synapse:syn3157743) and
syn25006650 (https://www.synapse.org/#!
Synapse:syn25006650) respectively. ROSMAP
transcriptomics and metabolomics raw data are
publicly available at Synapse under Synapse IDs
syn17008934 (https://www.synapse.org/#!
Synapse:syn17008934) and syn25878459 (https://
www.synapse.org/#!Synapse:syn25878459)
respectively. Information on how to access
controlled data (e.g., individual-level human data)
can be found at the AD Knowledge Portal (https://
adknowledgeportal.synapse.org/Data%20Access).
Uncontrolled data (e.g. processed data) can be
downloaded with only a Synapse account. Code
availability: The analysis codes used to integrate
cross-omics data for the three cohorts and all
downstream analyses were deposited in Zenodo
under the DOI: 10.5281/zenodo.10729969.

**Funding:** Research reported in this work was
supported by the Knight Alzheimer Disease
Research Center at Washington University School
of Medicine through the National Institute on Aging
(NIA: grant no. P30 AG066444 - JCM), Healthy
Aging and Senile Dementia (HASD: grant no.
P01AG003991 - JCM), and Antecedent Biomarkers
for Alzheimer Disease: The Adult Children Study
(ACS: grant no. P01AG026276 - JCM). This work
was supported by grants from the National
Institutes of Health: R01AG057777 (OH),
R01AG074012 (OH), U01AG072464 (OH),
K01AG046374 (CMK), R56AG067764 (OH),
R01NS118146 (BAB), R21NS127211 (BAB), and
K25AG083057 (AME) and by the Chan Zuckerberg
Initiative (CMK) and the BIDMC 2023 Translational
Research Hub Spark Grant Award (BAB). O.H. is an
Archer Foundation Research Scientist. A.M.E. is a
scholar recipient of the Knight ADRC Research
Education Component (NIA: grant no. P30
AG066444). The funders of the study had no role in

## Introduction

Alzheimer disease (AD) is a heterogeneous multifactorial neurodegenerative disorder patho-
logically characterized by amyloid (Aβ) plaques, neurofibrillary tangles (NFTs), neuroinflam-
mation, and synaptic and neuronal loss. Recently, distinct spatiotemporal trajectories of tau
pathology, brain atrophy, postmortem brain transcriptomics profiles, or cerebrospinal fluid
proteomics have been associated with multiple clinical and pathological AD features [1–10].
Differences in age at onset, sex, APOE ɛ4 carrier status, cognitive status, disease duration, and
cerebrospinal fluid (CSF) biomarker levels were associated with AD clinical progression and
neuropathological staging [1,5]. Molecular profiles associated with AD progression could
impact clinical trial design besides cognitive function, CSF, or imaging amyloid tau biomark-
ers. Novel biomarkers related to biological processes and cognition, in general, can shape new
clinical trials and are needed for evaluating the target engagement and efficacy of therapies tar-
geting amyloid plaques or tau tangles. However, the effect of AD molecular profiles on cogni-
tive decline remains poorly understood and understudied, representing a barrier to using
molecular classification effectively in clinical trials of potential disease-modifying therapies.

We have previously leveraged high-throughput brain molecular data from AD patients and
healthy individuals using transcriptomic [11–13], proteomic [14], metabolomic [15], and sin-
gle-cell omics [16] to study the signatures for AD. However, we recognize that omics layers are
interdependent and interconnected. Thus, we hypothesized that by combining multiple signals
from various molecules, novel significant biological hubs, or pathway changes in AD that are
otherwise missed with single-omic analyses will emerge. Previous studies that aggregate multi-
ple modalities of omics (cross-omics) have proven to be a powerful approach to identifying
molecular subtypes for complex human diseases including cancer [17–22]. Different modali-
ties of omics layers describe a biological system at different biomolecular levels [23], allowing
more comprehensive disease segregation into molecular profiles that cannot be appreciated
through an analysis restricted to a single modality or if modalities were analyzed separately.
Cross-omics approaches can reveal not only how complex biomolecular profiles change in
association with a disease, but also the relationships and correlations between distinct classes
of biological molecules. Their potential for studying neurodegeneration has been previously
described [24–34]. However, only recently has the deep molecular characterization of multiple
cortical regions from diverse brain banks and cohorts been generated [2,3], enabling a more
detailed and comprehensive examination of cross-omics data in AD. While several studies that
leveraged multiple layers of omics data were conducted in recent years and provided signifi-
cant insights into the pathological basis of AD [35–38], a few of them have fully utilized the
potential of cross-omics integration.

Previous studies have found molecular features associated with different AD clinical and
pathological profiles [2,3,10]. For example, CSF proteins for data-driven clustering found an
association between hyperplasticity and increased BACE1 levels with clinical differences in
AD patients [2]. Similarly, a system biology approach studying transcriptomics from multiple
cortical regions identified AD molecular subtypes associated with multiple dysregulated path-
ways including susceptibility to tau-mediated neurodegeneration, amyloid-β neuroinflamma-
tion, synaptic signaling, immune activity, mitochondria organization, and myelination [3].
Despite the substantial effort in these studies to identify molecular associations with clinical
and pathological AD features, broader conclusions are limited by its nature, as single-omic
analyses typically capture changes in a single component of the biological cascade. Additional
studies have leveraged cross-omics data integration approaches to study the alterations of
molecular and cellular pathways underlying AD pathophysiology [39,40] providing more evi-
dence that single-omic analyses do not capture most pathways involved in AD pathology.

the collection, analysis, or interpretation of data, in the writing of the report, or in the decision to submit the paper for publication.

**Competing interests:** The authors have declared that no competing interests exist.

**Abbreviations:** AD, Alzheimer disease; ADMC, Alzheimer's Disease Metabolomics Consortium; AOD, age of death; BIC, Bayesian information criteria; CDR, Clinical Dementia Rating; CSF, cerebrospinal fluid; DE, differential expression; DEG, differentially expressed gene; DLPFC, dorsolateral prefrontal cortex; DS, Down syndrome; FDR, false discovery rate; GLM, generalized linear model; IQR, interquartile range; LOD, limit of detection; MAPT, microtubule-associated protein tau; MCI, mild cognitive impairment; MCMC, Markov Chain Monte Carlo; MSBB, Mount Sinai Brain Bank; NCI, No Cognitive Impairment; ND, neurodegenerative disease; NFT, neurofibrillary tangle; OPC, oligodendrocyte precursor cell; PMI, postmortem interval; QC, quality control; RIN, RNA integrity number; ROSMAP, Religious Orders Study and Memory and Aging Project; snRNA-seq, single-nuclei RNA-seq; TCA, tricarboxylic acid; TMT, tandem mass tag.

Moreover, CSF multi-omics molecular signatures differentially related to AD pathology have also been identified [24]. Recently, integrating AD brain and blood multi-omics data with clinical and pathological data identified three molecular subtypes and inferred their respective molecular trajectories that diverge from the neuropathologically free control brains [10]. Furthermore, the molecular alterations underlying AD progression and heterogeneity are more clearly understood when studied in multiple brain regions with different burdens of AD pathologies. For example, the parietal cortex, an understudied brain region affected in later stages of AD [16,41,42], can better capture the more initial molecular changes in AD etiology compared to other severely affected regions (e.g., DLPFC), which will help to identify molecular dysregulations underlying AD progression before a higher burden of Aβ plaque and tangle occurs.

In this study, we sought to investigate whether cross-omics studies of AD brains can provide novel molecular insights into the progression of AD and its staging information that typical single-omic analyses cannot offer. We leveraged machine learning, digital deconvolution, and traditional statistical approaches to integrate and analyze multiple high-throughput *omics* datasets from different cortical regions including the parietal cortex, parahippocampal gyri (PHG, BM36), and dorsolateral prefrontal cortex (DLPFC) obtained from multiple AD cohorts. Our cross-omics data integration approach identified 4 distinct molecular profiles of AD, one of which was associated with worse cognitive function and neuropathological features including significantly higher Clinical Dementia Rating (CDR) at death, shorter survival after symptom onset, more severe neurodegeneration and astrogliosis, and decreased levels of metabolomic profiles. The molecular signatures of this profile, present in multiple cortical regions, show a significant dysregulation of synapse-related genes and pathways, suggesting neuron/synapse losses and dysfunction at later stages of AD. Integrating cross-omics approaches with data from established mouse models of neurodegeneration clarified the role of these genes and pathways in AD pathophysiology. Furthermore, combining cross-omics profiles identified in brains with single-nucleus resolution uncovered molecular profiles associated with AD features to unprecedented granularity and complexity. Subsequent AD staging, network analyses, and survival analyses of CSF proteins related to the synapses revealed their dysregulation associated with an increased risk of dementia progression, suggesting them as possible biomarkers for early synaptic dysfunction, cognition, and AD staging.

Our results suggest that cross-omics analyses capture molecular heterogeneity among brains from AD cases that are otherwise missed in non-clustered single-modality *omics* approaches and demonstrate that, under molecular stratification, cross-omics approaches provide new perspectives of molecular pathways associated with AD that are reflected in associations of cognitive decline progression and peripheral tissue. These novel molecular findings may open the possibility for new biomarkers for the molecular staging of AD and potential therapeutic targets to alleviate cognitive decline and AD progression that may provide a foundation for implementing precision medicine approaches for AD.

## Results

### Study design

We leveraged high-throughput transcriptomic (bulk RNA-seq), proteomic, metabolomic, and lipidomic data from the parietal cortex of participants with neuropathology-confirmed AD from the Knight Alzheimer Disease Research Center (Knight ADRC) for our discovery cohort (see Materials and methods, **Fig 1** and **S1 Table**). Omics data were generated using next-generation sequencing/10X genomics, SomaScan, and Metabolon Precision Metabolomics platforms for transcriptomics (60,0754 features) [11–13], proteomics (1,092 proteins) [14], and

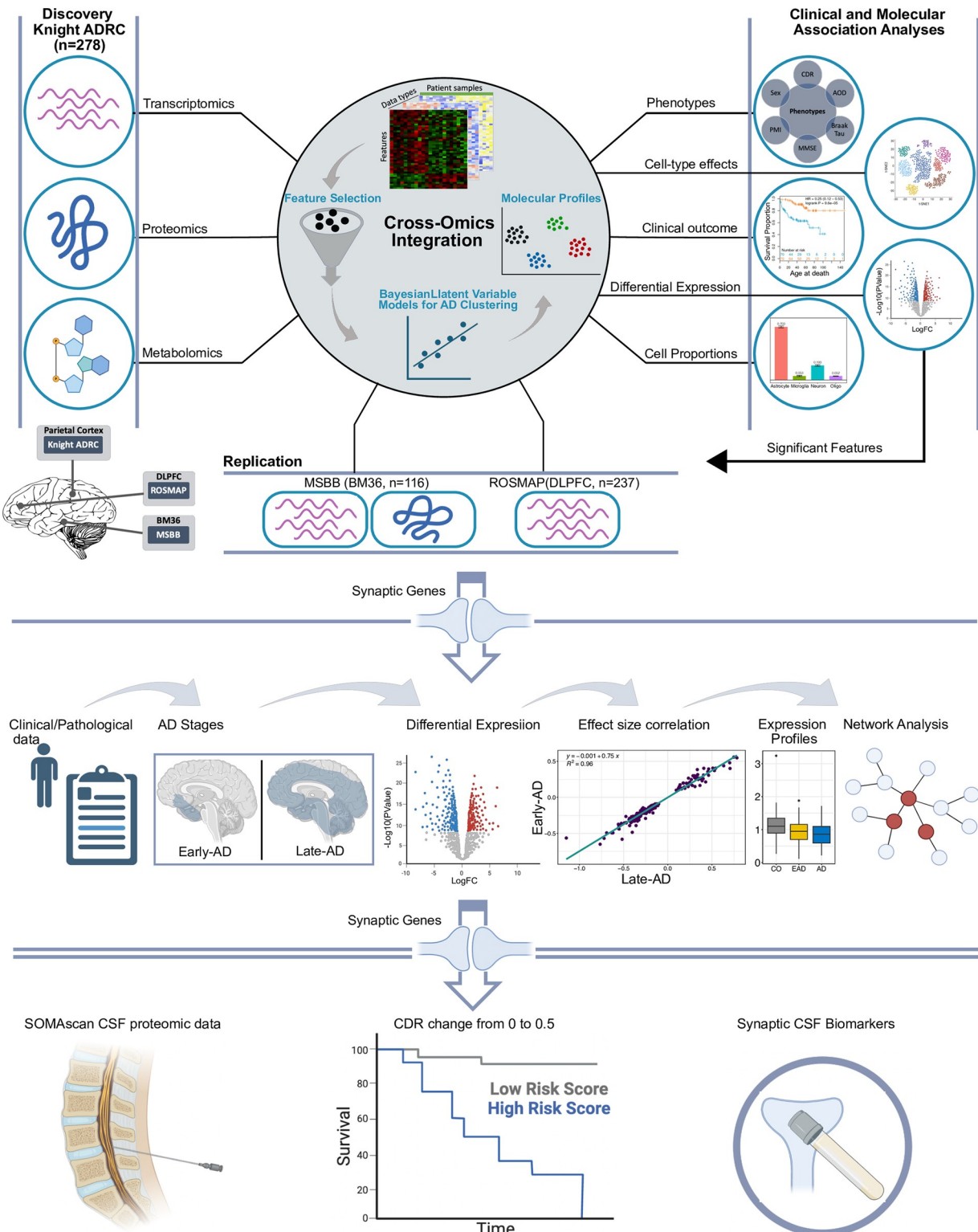

**Fig 1. Study design.** . The transcriptomics, proteomics, metabolomics, and lipidomics profiles of postmortem parietal cortex samples from the Knight ADRC participants were used. The 3 omics shared 278 subjects (255 sporadic AD and 23 control cases). Expression/reading matrices were prepared for the same set of samples (*n* = 278) and different numbers of features (features = 60,754, proteins = 1,092, metabolites = 627). Before the integration, feature selection was performed on all omics datasets whose number of features exceeded 1,000 by selecting the top and most variant features. A Bayesian integrative clustering method was then employed to integrate the 3 omics datasets and the best clustering

solution was extracted based on the maximum deviance ratio and the minimum BIC. AD molecular profiles were then linked to multiple clinical and molecular attributes to examine whether an association with these attributes exists. In addition, survival, DE, cell type-specific effect, and cell proportions inference analyses were performed on these profiles. These molecular profiles were then replicated in 2 independent datasets from MSBB BM36 (AD = 93, CO = 23) and ROSMAP (AD = 144, CO = 93). Dysregulated synaptic genes in the Knight-C4 profile were extracted and used for downstream AD staging analyses and in the ROSMAP cohort (early-AD vs. control and late-AD vs. control). Common synaptic genes present in the CSF SOMA dataset and associated with an increased risk of dementia through survival analyses were identified as CSF synaptic biomarkers for AD staging. Some components of this figure were Created with BioRender.com and edited and combined with Inkscape. AD, Alzheimer disease; BIC, Bayesian information criteria; CSF, cerebrospinal fluid; DE, differential expression; MSBB, Mount Sinai Brain Bank; ROSMAP, Religious Orders Study and Memory and Aging Project.

metabolomics (627 metabolites) [15], respectively (see Materials and methods). Preprocessing, quality control (QC), and feature selection were performed on each omics data type, and only samples that passed the QC criteria in all 3 omics modalities were retained, resulting in 278 samples (255 AD and 23 control). We employed a Bayesian integrative clustering method (iClusterBayes [43]) to integrate AD multi-omics data (see Materials and methods) and identify molecular profiles of AD. Control cases were later integrated for comparative analyses.

We next performed multiple association analyses including association with clinical and neuropathological attributes (e.g., sex, age of death, age at onset, postmortem interval (PMI), CDR, Braak amyloid stage, Braak neurofibrillary tangle stage), survival, and differential expression analyses to characterize AD molecular profiles. We leveraged digital deconvolution approaches to infer the cellular population structure using bulk RNA-seq data [13] and tested for association of these profiles with cell-type abundances. We also integrated single-nuclei RNA-seq data (snRNA-seq) from the parietal cortex of 67 Knight ADRC participants [16] to determine the specific cell-type effects.

For replication, we leveraged transcriptomics (bulk RNA-seq) and proteomics (tandem mass tag; TMT) data from the BM36 of 116 brains (93 AD and 23 control, S1 Table) from The Mount Sinai Brain Bank (MSBB) study [44] cohort, as well as transcriptomics data from 237 brains (144 AD and 93 control, S1 Table) DLPFC samples from The Religious Orders Study and Memory and Aging Project (ROSMAP) cohort [45], accessed through the AMP-AD Synapse portal. Preprocessing, QC, and feature selection were performed in the same way used to generate the discovery datasets.

Furthermore, we stratified AD cases into early and late-stage AD based on clinical and neuropathological rules from ROSMAP study (see Materials and methods, Fig 1) and performed DE and co-expression network analyses to identify the magnitude of the dysregulation of genes at distinct AD stages and provided a deeper understanding of the underlying dynamics of AD and approximate its progression over time (Fig 1). Similarly, we evaluated the association of protein levels encoded by selected genes in CSF and disease risk progression (CDR change from 0 to 0.5) using survival analyses in CSF proteomic data from the Knight ADRC participants (Fig 1).

## A multimodal profile of AD brains is associated with poor cognitive function and molecular attributes

We initially identified distinct AD molecular profiles by optimizing the number of clusters needed to capture the molecular heterogeneity present in AD brains from the Knight ADRC, independently of any neuropathological or clinical data (S1 Fig). We identified 4 clusters (Fig 2A) in which the molecular profiles of Cluster4 (named Knight-C4 hereafter, $n = 42$) exhibited a pronounced dysregulation of genes, proteins, and metabolites. For example, the transcriptomic profiles of the top 75% of quantile genes contributing significantly to the clustering solution show significant dysregulation signatures associated with Knight-C4 (Fig 2B).

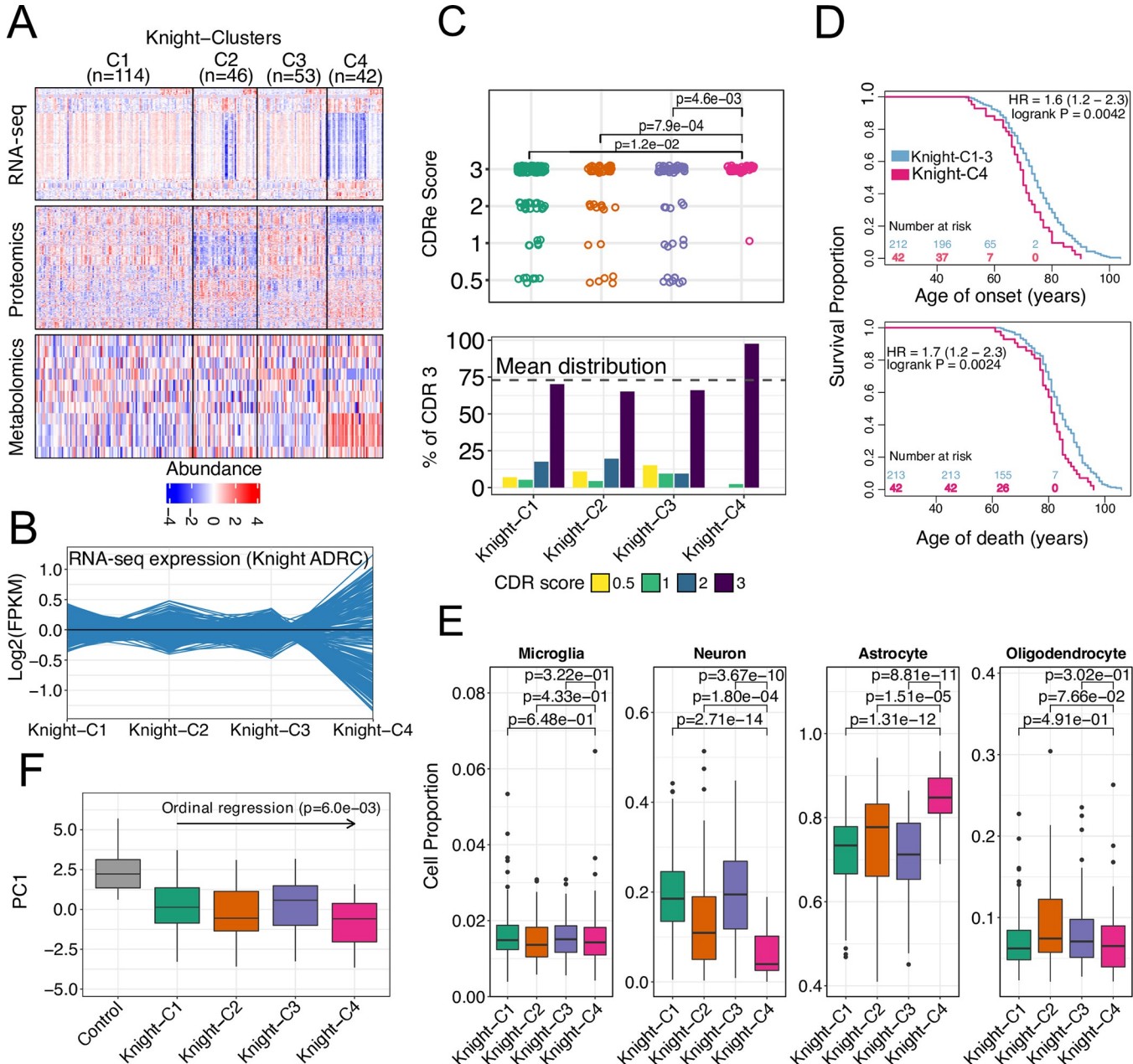

**Fig 2. Cross-omics data integration identified 4 distinct molecular profiles of AD associated with worse clinical outcomes and molecular attributes.** (A) Heatmaps of multi-omics profiles of the top significant features. (B) Transcriptomic profiles (scaled) of the top genes contributed significantly to the clustering, showing an overall dysregulation of brains in Knight-C4. (C) The distribution of CDR scores across 4 clusters with Knight-C4 associated with high CDR scores. (D) Kaplan–Meier plots showing Knight-C4 association with poor outcome for the age of onset and death. (E) Boxplots showing the cell proportion from deconvolution analysis across 4 clusters using bulk RNA-seq from 4 cell types. Knight-C4 is associated with a significantly higher and lower proportion of astrocytes and neurons, respectively. (F) Boxplot showing each cluster's first principal component of metabolomics profiles, with Knight-C2 and Knight-C4 showing a lower metabolomic signature than other AD cases. The data underlying panels C, E, and F can be found in **S1** Data. AD, Alzheimer disease; CDR, Clinical Dementia Rating.

We then evaluated whether these molecular profiles were associated with genetic risk factors, neuropathological status, or clinical and demographic variables. We found that AD cases in Knight-C4 showed significantly higher Clinical Dementia Rating (CDR) scores at death ($p = 5.2 \times 10^{-3}$; **Fig 2C**), shorter survival time following disease onset (8 months shorter,

**Fig 2D**, $p = 4.2 \times 10^{-3}$, HR = 1.6), and younger age at death (**Fig 2D**, $p = 4.2 \times 10^{-3}$, HR = 1.7) compared to AD cases in Knight-C1-3. The Knight-C4 cluster was not associated with either AD polygenic risk score (**S2 Fig**), *APOE* ε4 allele carrier status (**S3A Fig**), or Braak stages (neurofibrillary tangle or amyloid) (**S3B and S3C Fig**). However, Knight-C4 included more Braak tau stage VI cases than the other clusters (**S3B Fig**). The parietal cortex is affected in later stages of AD, which could explain the lack of correlation of Knight-C4 cluster with Braak staging.

Next, we inferred the cellular population structure from bulk RNA-seq [13] leveraging digital deconvolution methods (see Materials and methods). Knight-C4 showed significantly higher astrocytes and lower neuronal proportions (**Fig 2E** and **S2 Data**). We then evaluated whether a metabolomic profile that we previously reported to be associated with AD duration [15] would reveal additional molecular differences among these clusters. Lower values of this metabolomic profile (or *eigenmetabolite*) were associated with a longer disease duration driven by earlier onset [15]. Knight-C2 and Knight-C4 exhibited significantly lower eigenmetabolite scores (**Fig 2F**) using ordinal logistic regression ($p = 6.0 \times 10^{-03}$). Our approach identified new AD molecular profiles linked to cognitive test scores, the pace of disease progression, survival with the disease, the proportion of neuronal loss, astrocytosis, and metabolic changes.

## Knight-C4 profile is replicated consistently across multiple independent cohorts and brain regions

Our analyses of the transcriptomics and proteomics data from AD parahippocampal gyrus (BM36) from the MSBB study [44] (**S1 Table**) identified 2 distinct molecular profiles (**Fig 3A**). Cluster1 (MSBB-C1) recapitulated the transcriptomic signatures identified in Knight-C4. We also ascertained the extent of overlap between the molecules dysregulated in Knight-C4 (**S3 Data**) and MSBB-C1 (**S4 Data**) and determined a coincident profile of dysregulated genes (**Figs 3G–3I** and **S4A**, $p < 2.2 \times 10^{-16}$ hypergeometric test and **S5 Data**). Furthermore, MSBB-C1 also exhibited more severe dementia ($p = 1.5 \times 10^{-3}$, **Fig 3B**) and estimated higher and lower proportions of astrocytes ($p = 5.5 \times 10^{-07}$) and neurons ($3.2 \times 10^{-09}$), respectively (**Fig 3C** and **S2 Data**) compared to other AD cases in MSBB. MSBB-C1 was also associated with higher Braak tau scores ($p = 3.9 \times 10^{-03}$, **S5A Fig**). Although not significant ($p > 0.05$), these donors show a trend suggesting earlier age of death (**S5B Fig**). We did not identify a significant overlap of dysregulated proteins between regions ($p > 0.05$, **S4B Fig**). We also observed that the molecular profile of MSBB-C1, first detected in the parahippocampal gyrus (BM36), is consistently present in multiple cortical regions including the frontal cortex (BM10), superior temporal gyrus (BM22), and inferior frontal gyrus (BM44) (**S5C Fig**).

Likewise, our analyses of the RNA-seq data from the DLPFC from the ROSMAP cohort [45] identified a similar subset of AD brain donors (**Fig 3D**). Relative to those in Cluster2 (ROSMAP-C2), Cluster1 (ROSMAP-C1) recapitulated the transcriptomic profiles of those in Knight-C4 (**Figs 3G–3I** and **S4C,** $p < 2.2 \times 10^{-16}$ hypergeometric test and **S5 and S6** Data). They also showed higher tau tangle densities ($p = 1.3 \times 10^{-02}$) and neurofibrillary tangle burdens (PHF tau tangles, $p = 2.1 \times 10^{-02}$; **S6A and S6B Fig**), significantly higher astrocyte ($p = 3.1 \times 10^{-08}$) and lower neuron ($p = 9.7 \times 10^{-14}$) proportions (**Fig 3F** and **S2 Data**), and a nonsignificant ($p > 0.05$), trend to higher plaque densities—Global burden of AD pathology (**S6C Fig**, see Materials and methods and Bennett and colleagues [45] for details on how these neuropathological variables are identified). Some of these findings were also presented in [10]. ROSMAP-C1 also showed evidence of greater cognitive impairment and increased neuropathological AD scores (MMSE; $p = 4.4 \times 10^{-02}$; **Fig 3E**). Advanced AD patients with poor cognitive function show similar cellular and molecular profiles in multiple affected cortical regions.

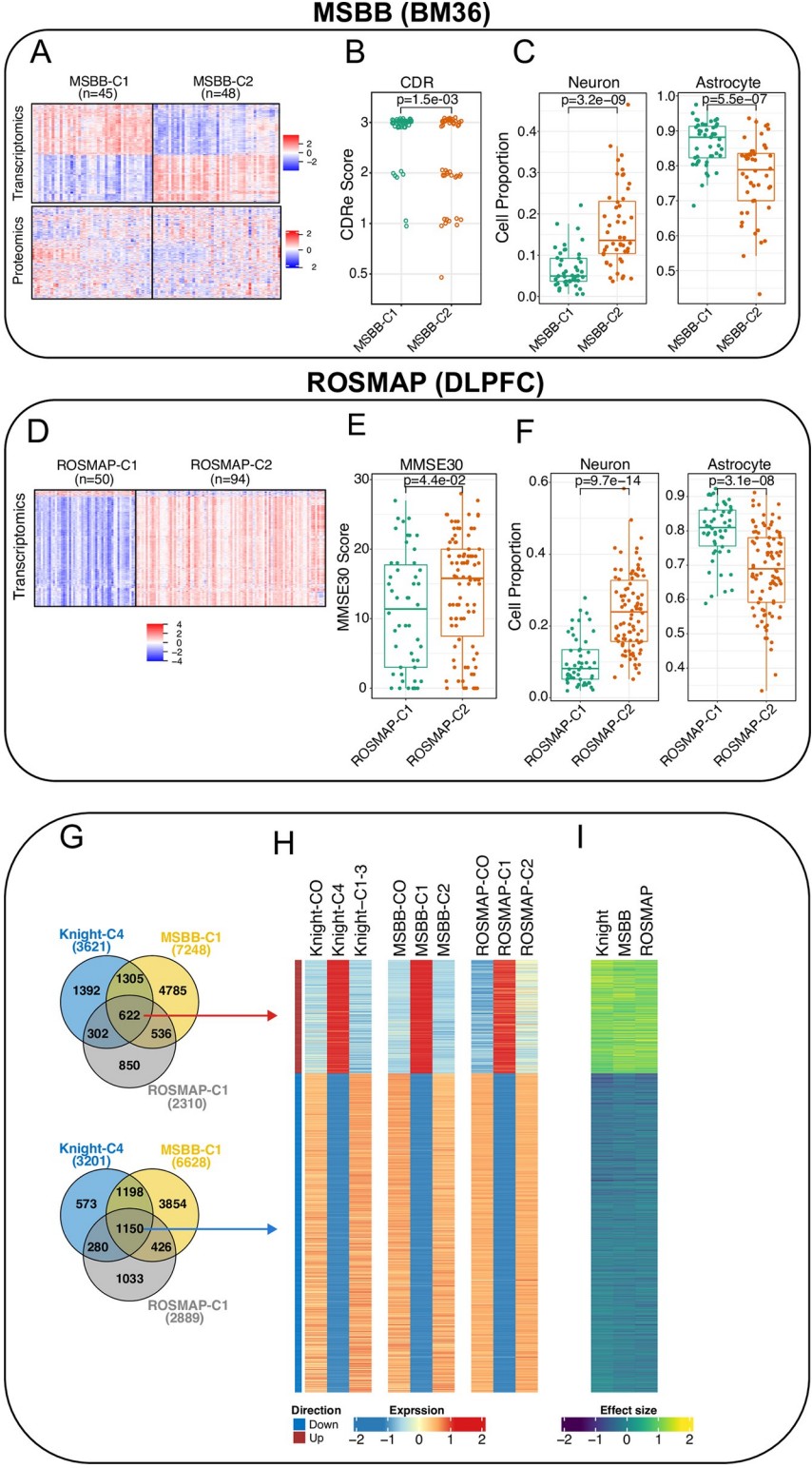

**Fig 3. Molecular profiles of Knight-C4 are replicated in 2 independent datasets.** (A) Heatmaps of the transcriptomic and proteomic profiles of the top features from MSBB (BM36) show 2 distinct clusters. (B) Boxplots showing MSBB-C1 associated with higher CDR scores, replicating Knight-C4 in the knight ADRC. (C) Boxplots showing cell proportions inferred from bulk RNA-seq from MSBB (BM36) using digital deconvolution. MSBB-C1 replicates Knight-C4 by showing an association with significantly higher and lower proportions of astrocytes and

neurons, respectively. (D) Heatmap of the transcriptomic profiles of the top features from ROSMAP (DLPFC) showing 2 distinct clusters. I Boxplots showing ROSMAP-C1 associated with lower MMSE30 scores. (F) Boxplots showing cell proportion estimated from bulk RNA-seq from ROSMAP (DLPFC) using digital deconvolution. Like MSBB, ROSMAP-C1 replicates Knight-C4 by showing an association with significantly higher and lower proportions of astrocytes and neurons, respectively. (G) Venn diagram showing the common dysregulated genes in the discovery and replicated cohorts. (H) Heatmaps of the mean expression of the shared genes across the discovery and replicated cohorts showing a clear, consistent expression pattern. (I) Heatmap of the effect size detected in the discovery and replicated cohorts showing a high effect size similarity across the 3 cohorts. The data underlying panels B, C, E, and F can be found in **S1** Data. CDR, Clinical Dementia Rating; DLPFC, dorsolateral prefrontal cortex; MSBB, Mount Sinai Brain Bank; ROSMAP, Religious Orders Study and Memory and Aging Project.

The molecular profiles associated with the severity of AD pathology can still vary even in the advanced stages of the disease.

## The Knight-C4 subjects exhibited a remarkable molecular disarray

Differential expression (DE) analyses (see Materials and methods, S3 Data) showed that Knight-C4 has approximately 10-fold increase in the number of significantly differentially expressed genes (DEGs) relative to controls, in comparison to Knight-C1-C3 (S7 Data and S7A Fig); furthermore, 90% of the 4,661 DEGs were unique to participants in the Knight-C4 (S7 Data and Fig 4A, green bars). Similarly, we observed many proteins and metabolites with significantly different abundance levels in Knight-C4 relative to controls (S8 and S9 Data). However, Knight-C2 has the largest dysregulated proteins (S7 Data and S7B and S7C Fig). We identified 24 (34%), 232 (51%), and 250 (52%) significant proteins unique to Knight-C1, 2, and 4, respectively (Fig 4B, green bars and S7 Data). We did not detect significant proteins that were dysregulated uniquely in Knight-C3 (S7B Fig). Knight-C4 was associated with many cluster-specific metabolites significantly different from controls (90 hits; S7 Data and S7C Fig), whereas only 1 cluster-specific metabolite significantly different from controls was associated with Knight-C2. Of the 90 hits, 80 (90%) significant metabolites significantly differed between Knight-C4 and other AD cases.

We also performed DE analyses comparing all AD cases combined (unclustered) to control brains. We determined the number of significant cluster-specific features not captured by unclustered analyses (S7 Data and Fig 4A–4C, purple bars). We observed that Knight-C4 exhibited the highest number of features missed by unclustered analyses with 4,295 genes (92%), 475 proteins (98%), and 90 metabolites (100%) (S7 Data and Fig 4A–4C, purple bars), indicating the importance of clustered analyses. Notably, a high percentage of features was also observed with Knight-C2 in proteomics (98%). In contrast, unclustered analyses identified a small percentage of significant features not captured by clustered analyses. For example, only 5% and 0.5% of unclustered significant features for transcriptomics and proteomics, respectively, were not captured by Knight-C4. Knight-C4 missed no unclustered significant metabolites.

DE and abundance analyses comparing brains in each AD cluster to brains in other AD clusters (S3, S8, and S9 Data) revealed that Knight-C4 showed the highest percentage of significant hits compared to other AD brains (Fig 4D and S10 Data). This significant difference was observed in transcriptomics, with 6,822 significant hits compared to 678, 22, and 1 for Knight-C1-3 (Fig 4D, outer track and S10 Data). Similarly, this difference was observed in metabolites with 228 significant hits in Knight-C4 compared to 105, 0, and 0 for Knight-C1-3, respectively (Fig 4D, inner track and S10 Data). No significant differences in the number of hits were observed between the 4 clusters in proteomics, with all clusters showing relatively similar detections (Fig 4D, middle track and S10 Data). We also verified that none of the

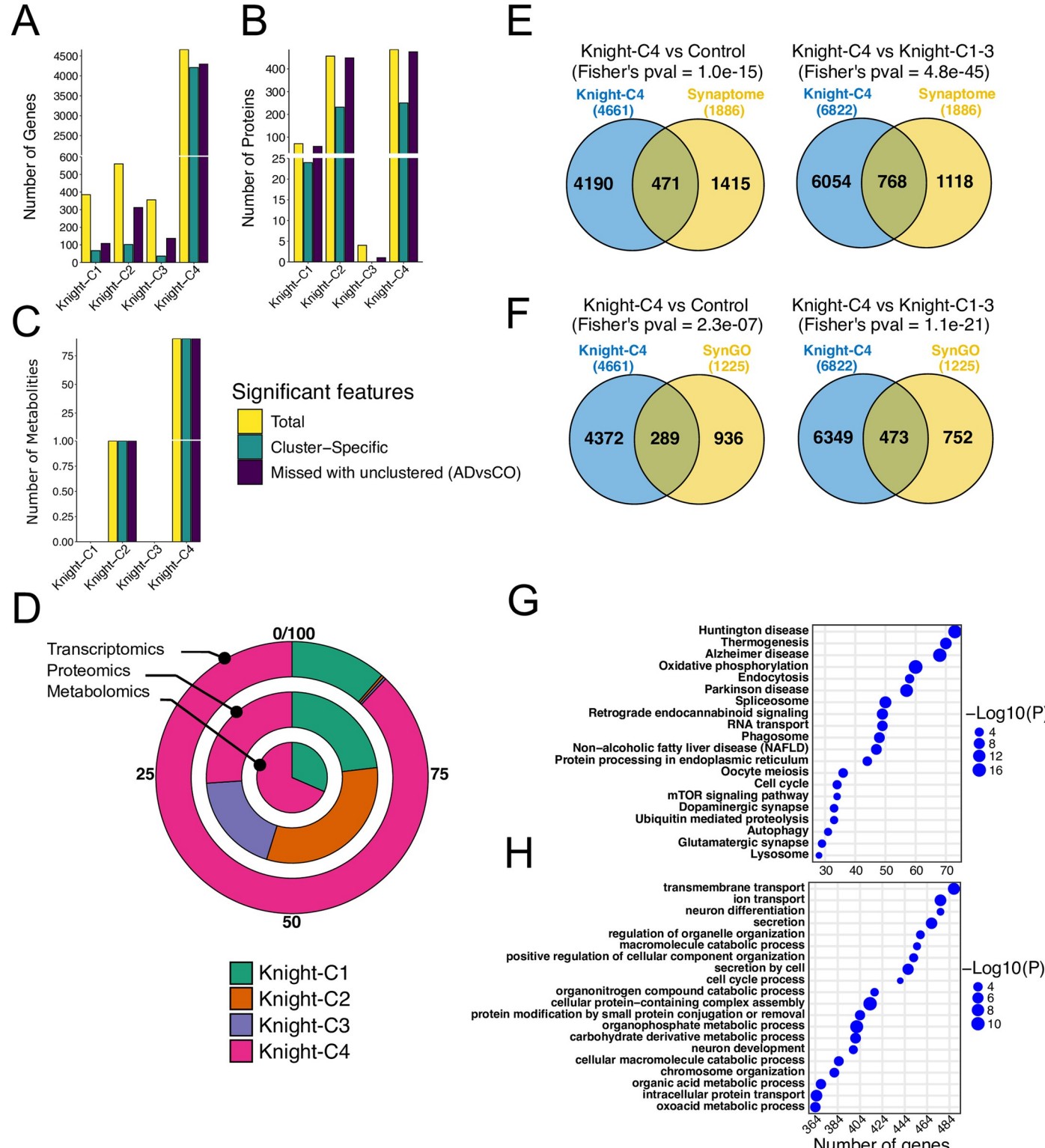

**Fig 4. The parietal cortex of participants with worse cognitive function exhibited remarkable molecular dysregulation.** (A) Bar plots showing the significant genes detected in each cluster compared to the control. Yellow represents the total number of genes, green represents the cluster-specific genes, and purple depicts the number of genes missed with an unclustered approach (all AD cases vs. control). (B) Same as panel "A" but for proteins. (C) Same as panel "A" but for metabolites. (D) Race track plot showing the percentage of significant features for each cluster compared to other clusters for the 3 omics. (E) Venn diagrams showing the overlap between significant genes in Knight-C4 and synaptic genes ("synaptosome") from Fei and colleagues [46]. (F) Same as "E" but overlaps with

SynGO dataset [47]. The significance of overlap was computed using Fisher's exact test. (G) Top 20 KEGG pathways associated with Knight-C4. (H) Top 20 GO biological process pathways associated with Knight-C4. The data underlying panels A, B, and C can be found in S1 Data. AD, Alzheimer disease.

omics dominate this solution, each capturing molecular heterogeneity among AD brains that are not necessarily detected by other *omics* modalities.

Noticeably, these DE analyses also showed that Knight-C4 is significantly enriched in dysregulated synaptic genes (**Fig 4E** and **4F** and **S11 Data**) using the annotation from The SynGO consortium [47] ($p = 1.0 \times 10^{-15}$ and $p = 4.8 \times 10^{-45}$ for Knight-C4 versus Control and Knight-C4 versus Knight-C1-3 sets, respectively) and curated list from Fei and colleagues [46] ($p = 2.3 \times 10^{-07}$ and $p = 1.1 \times 10^{-21}$ for Knight-C4 versus Control and Knight-C4 versus Knight-C1-3 sets, respectively). MSBB-C1 and ROSMAP-C1 profiles were also significantly enriched in synaptic genes (**S8A** and **S8B Fig** and **S11 Data**).

In summary, these results show that molecular profiles of AD are heterogeneous, and the cross-omics integration approach can better identify the substantial differences between clinically relevant subcategories of AD brains.

## The Knight-C4 cluster is significantly enriched in AD-related dysregulated pathways

Knight-C4 contains multiple pathways related to different aspects of neurodegeneration (**Fig 4G** and **4H**). For example, KEGG pathways identified multiple intracellular trafficking and signaling pathways (**Fig 4G**) including endocytosis ($p = 6.5 \times 10^{-04}$; notable genes, *VPS29* and *VPS35*), phagosome ($p = 1.2 \times 10^{-06}$; notable genes, *ATP6V1A* and *RAC1*), and mTOR signaling pathway ($p = 3.3 \times 10^{-02}$; notable genes, *GSK3B*, and *IGF1*), all of which have been shown to play major roles in the pathological processes of AD [48–50]. Genes (*PPT1*, *GBA*, *HEXB*, and *LGMN*) from the lysosomal pathway ($p = 4.7 \times 10^{-02}$), which is critical for microglia, and neurons were also identified (**Fig 4G**). Growing evidence has shown that the dysfunction of lysosomes plays a major role in the pathogenesis of multiple neurodegenerative diseases (NDs) including the progression of AD [51–55]. Knight-C4 was also associated with genes (*BECN1* and *MAPK1*) from the autophagy pathway ($p = 1.6 \times 10^{-02}$, **Fig 4G**), which is associated with several neurodegenerative disorders including AD [56–59]. For example, autophagy dysfunction was shown to be linked to the accumulation of misfolded protein aggregates [60]. The down-regulation of Knight-C4 compared to other AD cases identified multiple ND-related pathways (**Fig 4G**) including AD ($p = 2.5 \times 10^{-14}$), Huntington disease ($p = 3.3 \times 10^{-14}$), and PD ($p = 2.1 \times 10^{-12}$). Knight-C4 also exhibited down-regulation of synaptic pathways including dopaminergic synapse ($p = 7.0 \times 10^{-03}$) and glutamatergic synapse ($p = 1.0 \times 10^{-02}$) (**Fig 4G**) including *GRIA1*, *GRIA2*, *PPP3CA*, *GRIN2A*, *PLCB1*, *CALY*, and *GLS* associated genes. Furthermore, GO biological process analyses identified significant neuronal differentiation ($p = 1.8 \times 10^{-02}$) and development ($p = 5.7 \times 10^{-03}$) associated with Knight-C4 (**Fig 4H**). Most of these pathways were also identified in MSBB-C1 and ROSMAP-C1 (**S9A–S9C Fig**).

Compared to the control group, Knight-C4 showed additional pathways including herpes simplex virus 1 infection ($p = 4.4 \times 10^{-03}$), a risk factor for AD [61–64] that lead to neuronal damage and induced gliosis [65,66], and synaptic vesicle cycle ($p = 4.6 \times 10^{-05}$) shown to play a key role in the pathobiology of AD [67] (**S9D Fig**). Noticeably, herpes simplex virus 1 infection was significantly enriched in Zinc Finger genes, a class of protein-coding genes known to regulate gene expression at the transcriptional and translational levels by binding to specific DNA or RNA molecules. GO biological process analyses identified several significant synaptic

signaling and transmission pathways including 2 well-known AD biomarkers, *SNAP25* and *SYT1* (**S9E Fig**). Many of these pathways were also present in MSBB-C1 and ROSMAP-C1 (**S10A–S10D Fig**). Pathway analyses using significant proteins identified AD-related pathways including PI3K−Akt signaling ($p = 6.5 \times 10^{-19}$), rap1 signaling ($p = 5.8 \times 10^{-14}$), axon guidance ($p = 2.1 \times 10^{-12}$), MAPK signaling ($p = 1.4 \times 10^{-20}$), and ras signaling ($p = 1.8 \times 10^{-15}$) pathways associated with the down-regulation of Knight-C4 (**S10E Fig**). Noticeably, *MAPT*, the gene that encodes the microtubule-associated protein tau (MAPT), was among the genes in the MAPK signaling pathway. Top GO pathways, enriched in Knight-C4 down-regulated proteins (and replicated in MSBB-C1), were related to cell/gene regulatory pathways (**S10F Fig**). Through cross-omics analyses, we discovered numerous known and novel pathways contributing to the cognitive decline in AD patients in multimodal profiles.

## The multimodal clusters identify candidate AD biomarkers

Among the top genes/proteins dysregulated in Knight-C4, several AD biomarkers including *APP*, *APOE*, *CLU*, *SNAP25*, *GFAP*, *SNCA*, *NOTCH3*, *TARDBP*, *GRN*, *MMP9*, and *C9ORF72* were identified (**S3**, **S8,** and **S9 Data** and **Fig 5A** and **5D**). Of these, transcript levels of Synaptosome Associated Protein 25 (*SNAP25*), a protein implicated in AD and Down syndrome (DS) [68], were significantly down-regulated in Knight-C4 (**Fig 5B**), as it has been previously reported in AD [68]. *SNAP25* protein levels were also shown to be reduced in the CSF in AD patients [69]. *SNAP25* was significantly DE between Knight-C4 and control ($p = 5.9 \times 10^{-3}$) and between Knight-C4 and other AD cases ($p = 4.2 \times 10^{-08}$, **Fig 5B**). However, *SNAP25* was not significantly DE ($p > 0.05$, **Fig 5B, left panel**) in unclustered AD cases compared to the control. No significant differences were observed in *SNAP25* protein levels in clustered or unclustered analyses. The decreased expression of *SNAP25* in Knight-C4 was replicated in MSBB-C1 (transcript and protein levels) and ROSMAP-C1 (transcript levels) cohorts (**S11A–S11C Fig**).

Glial Fibrillary Acidic Protein (*GFAP*), an emerging fluid biomarker in brain disorders [70] usually employed to identify astrocytic reactivity state, a well-documented process in AD and neurodegeneration, was identified. The increased astrocyte reactivity and overexpression of *GFAP* have been reported in multiple studies [68,70–78]. *GFAP* was overexpressed uniquely in Knight-C4 compared to Knight-C1-3 (**Fig 5C**, $p = 1.7 \times 10^{-2}$). However, *GFAP* was not significantly DE in Knight-C4 compared to controls, nor was it DE in all AD cases compared to the control ($p > 0.05$, **Fig 5C**). We repeated these analyses using a simplified model that did not correct for cellular population structure to examine whether collinearity between case-control status and cellular population structure may introduce spurious results, but we could not obtain a significant association. The same pattern was observed with *AQP4*, another marker gene of astrocyte reactivity state; it was only significantly different between Knight-C4 and Knight-C1-3 ($p = 1.0 \times 10^{-02}$). We did not identify significant differences in *GFAP* protein levels ($p > 0.05$, **S12 Fig**). However, protein levels of *GFAP* were significantly up-regulated in Knight-C2 (**S12 Fig**) compared to the control and AD brains in Knight-C1,3,4 ($p = 1.0 \times 10^{-4}$ and $1.4 \times 10^{-21}$, respectively). The increased transcriptomic *GFAP* levels were also identified in MSBB-C1 and ROSMAP-C1 (**S13A–S13C Fig**) but with less significance in ROSMAP-C1. Altogether, these results support heterogeneity in the presence of astrocytic reactivity state and its association with different AD subtypes.

Our cross-omics integration analyses also identified cluster-specific dysregulation of Clusterin (*CLU*), a gene associated with AD [79,80]. Genome-wide association studies have shown that *CLU* (*APOJ*) is the third most associated late-onset AD (LOAD) risk gene after *APOE* and *BIN1* [80–83]. It was also shown to play a neuroprotective role in AD by altering Aβ

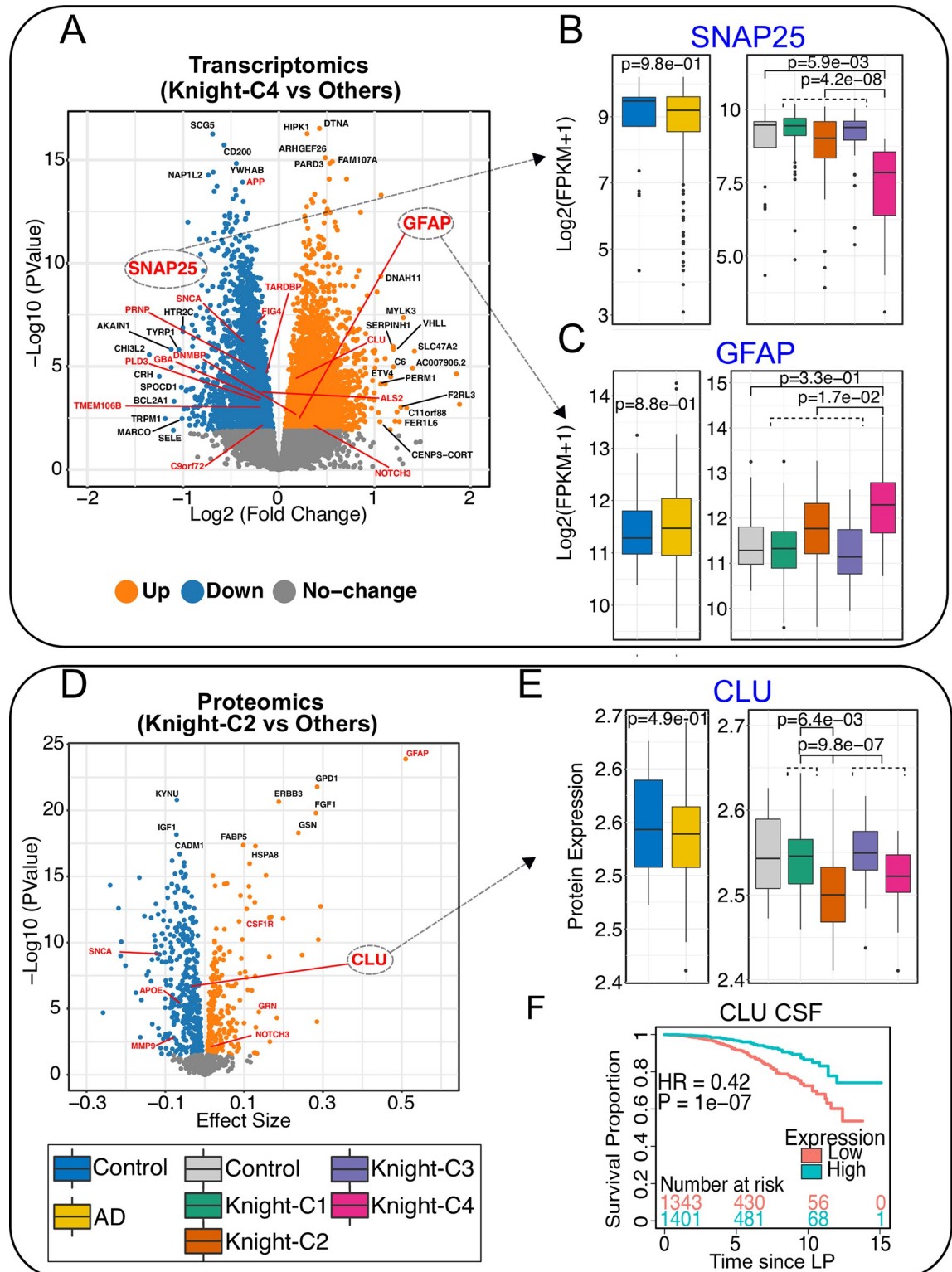

**Fig 5. DE analyses identified ND genes associated with distinct clusters.** (A) Volcano plot showing the up- and down-regulated genes identified between Knight-C4 vs. other clusters. Genes in red are examples of known ND genes. Genes in black are the top 10 significant genes based on the adjusted *p*-value and genes with log2(fold change) > 1. (B) Boxplots showing the transcriptomic profiles (in FPKM) of *SNAP25* across 4 clusters (right) and for AD cases combined (left) in the Knight ADRC cohort. (C) Same as "B" but for *GFAP*. (D) Volcano plot showing the up-and down-regulated proteins identified between Knight-C2 vs. other clusters. (E) Boxplots

showing the proteomic profiles of *CLU* across 4 clusters (right) and for all AD cases (left) in the Knight ADRC cohort. (F) Kaplan–Meier plot showing the association of Clusterin (ApoJ) protein concentrations in CSF with an increased risk of dementia progression using CDR change from 0 to 0.5 using the CSF dataset from the Knight ADRC participants. The data underlying panels B, C, and E can be found in **S1** Data. AD, Alzheimer disease; CDR, Clinical Dementia Rating; CSF, cerebrospinal fluid; DE, differential expression; ND, neurodegenerative disease.

aggregations and promoting Aβ clearance [83–87]. *CLU* protein levels were significantly down-regulated in Knight-C2 compared to the control ($p = 6.4 \times 10^{-3}$) and AD cases in Knight-C1,3,4 ($p = 9.8 \times 10^{-7}$; **Fig 5E**). *CLU* was also down-regulated considerably in Knight-C4 compared to the control ($p = 3.4 \times 10^{-02}$; **Fig 5E**). We did not observe significant differences in *CLU* transcriptomic and proteomic levels in the MSBB and ROSMAP cohorts. To assess the clinical implications of *CLU* dysregulation, we evaluated its levels in the CSF from 553 participants from the Knight ADRC by performing survival analyses for progression using CDR change from 0 to 0.5. These analyses indicate that lower CSF *CLU* is associated with an increased risk of dementia progression (HR = 0.42; $p = 1.0 \times 10^{-07}$, **Fig 5F**). Together, results show that the cross-omics approach identified potential biomarkers for AD that traditional approaches may miss.

## Metabolic differential abundance and pathway analyses in Knight-C4

We next looked at the significant metabolites that uniquely characterize brain samples in Knight-C4. To determine whether significant metabolites are replicated in additional cohorts, we accessed metabolomics from the same ROSMAP cohort [45] (**S1 Table**, Materials and methods). We performed differential abundance analyses comparing the levels from donors in the previously identified ROSMAP-C1 cluster to controls and other AD cases (**S6 Data**). We identified 57 significant metabolites (43 increased and 14 decreased) in Knight-C4 compared to controls replicated in the ROSMAP-C1 (**Fig 6A** and **S12 Data**). Similarly, we identified 74 metabolites (70 increased and 4 decreased) in the Knight-C4 compared to other AD cases replicated in ROSMAP-C1 (**Fig 6B** and **S12 Data**).

Pathway analyses using these common metabolites identified several pathways significantly related to AD. Among the metabolites decreased in AD (**Fig 6A** and **6B**), gamma-aminobutyrate (GABA) and choline, which involved major neurotransmitter pathways, were identified. GABA, synthesized in neurons, was significantly decreased in Knight-C4 ($p = 5.1 \times 10^{-04}$) and ROSMAP-C1 ($p = 1.3 \times 10^{-02}$) compared to the control (**S14A Fig**). The same observation was found when these profiles were compared to other AD cases ($p = 1.2 \times 10^{-05}$ and $p = 1.6 \times 10^{-02}$ for the Knight-C4 and ROSMAP-C1, respectively, **S14A Fig**). As an inhibitory neurotransmitter, GABA has been shown to play a key role in synchronizing the activity of the human cerebral cortex [88,89]. Similarly, the metabolomics profiles of choline were decreased in these profiles in both Knight-C4 ($p = 4.1 \times 10^{-06}$) and ROSMAP-C1 ($p = 1.2 \times 10^{-04}$) compared to controls (**S14B Fig**). These reading levels of choline also show significantly different abundance in these clusters compared to other AD cases ($p = 2.9 \times 10^{-07}$ and $p = 4.7 \times 10^{-3}$ for Knight-C4 and ROSMAP-C1, respectively, **S14B Fig**). Choline was also shown to have a critical role in neurotransmitter function because of its effect on the function of acetylcholine and dopamine [90]. However, we did not find significant differences in the reading levels of these active neurotransmitters. Nor did we observe significant differences for additional neurotransmitters such as serotonin and glutamate. We then studied tryptophan, an amino acid essential for metabolites that serve as neurotransmitters and signaling molecules [91]. We found that its levels increased in the Knight-C4 compared to the control and other AD cases ($p = 8.3 \times 10^{-3}$ and $3.0 \times 10^{-06}$, respectively) (**S15A Fig**). Similarly, we found that brains in ROSMAP-C1

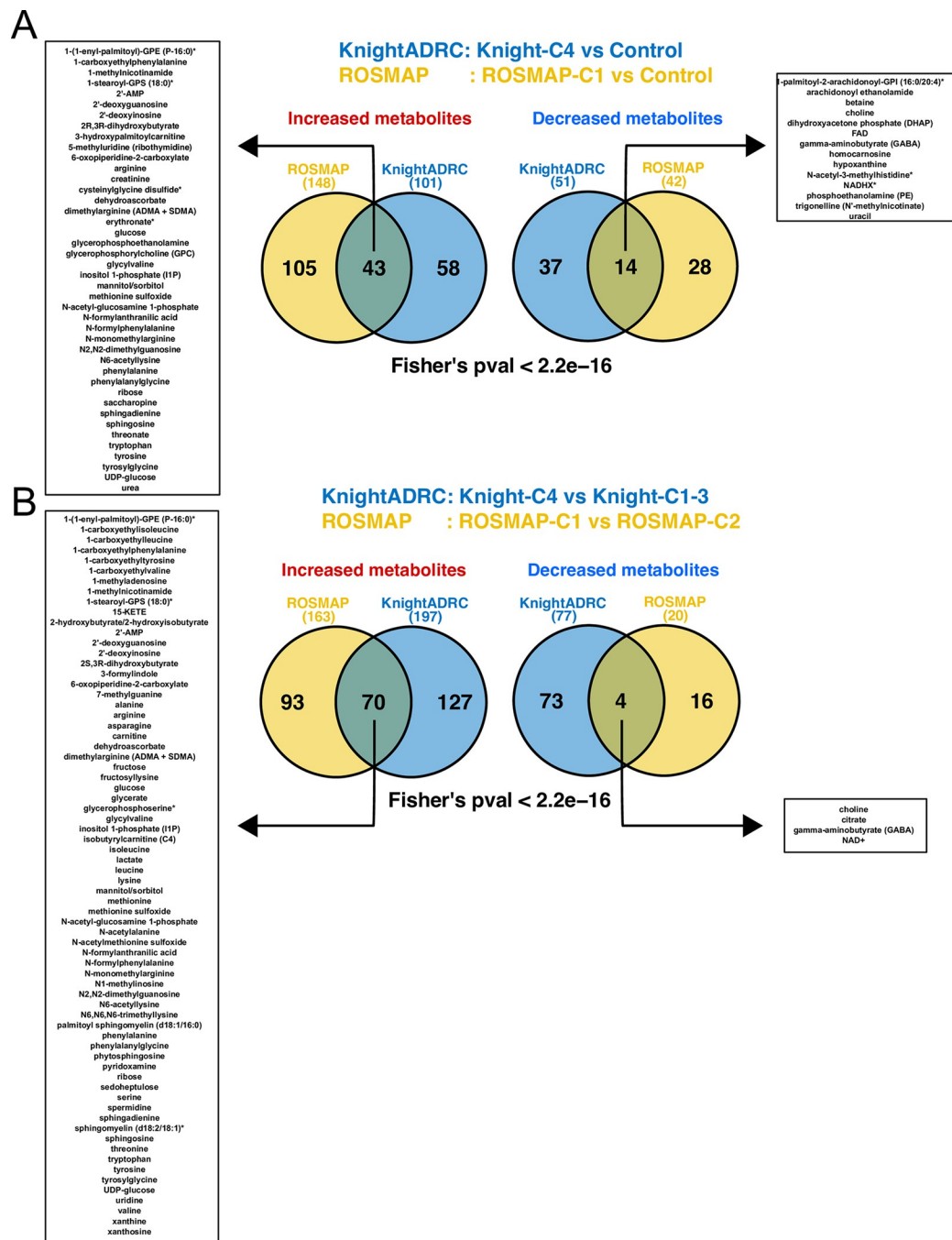

**Fig 6. Differential abundance analyses identified significant metabolites and pathways associated with Knight-C4.**
(A) Venn diagrams show the significantly increased and decreased metabolites shared between the Knight-C4 and ROSMAP-C1 compared to the control. The left and right boxes show the names of shared metabolites. (B) Similar to panel "A" but for metabolites shared between Knight-C4 and ROSMAP-C1 compared to the other cases. The significance of overlap was computed using Fisher's exact test. ROSMAP, Religious Orders Study and Memory and Aging Project.

showed increased tryptophan levels compared to the control and other AD brains ($p = 1.4 \times 10^{-4}$ and $2.8 \times 10^{-07}$, respectively) (**S15A Fig**).

Several increased carbohydrate/sugar metabolites were identified in Knight-C4 (**Fig 6A** and **6B**) including glucose, mannitol, sorbitol, ribose, and UDP-glucose. For example, glucose, a

metabolite implicated in the pathogenesis of AD [92,93], was significantly increased in Knight-C4 and ROSMAP-C1 (S15B Fig). Components of glycolysis and the tricarboxylic acid (TCA) cycle were also dysregulated. Specifically, levels of citrate were decreased in both Knight-C4 ($p = 9.7 \times 10^{-05}$) and ROSMAP-C1 ($p = 1.4 \times 10^{-03}$) compared to other AD cases (S16A Fig). NAD+, an important oxidizing cofactor in glucose metabolism and cellular respiration, was also decreased in both Knight-C4 ($p = 2.2 \times 10^{-02}$) and ROSMAP-C1 ($p = 2.2 \times 10^{-02}$) compared to other AD cases (S16B Fig). NAD+ is synthesized from tryptophan, and levels of NAD+ have been shown to decrease with age, potentially contributing to the pathogenesis of AD [94]. These observations align with previous findings that associate increased brain glucose and slowed glucose metabolism with more severe AD [93,95]. In addition, several increased sphingolipid metabolites were identified including sphingomyelin, sphingosine, and sphingadienine (Fig 6A and 6B). Previous studies showed the implication of sphingolipid metabolism deregulation in AD [96]. For example, sphingosine, which had higher levels in AD brains [96], showed significantly increased metabolism levels in Knight-C4 and ROSMAP-C1 (S16C Fig). These results show that cross-omics methods can uncover dysregulated metabolites in AD brains, indicating a link to other conditions like diabetes [95].

## Single-nucleus transcriptomics suggests neurons and alternative glial cells mediate cross-omics profiles of AD

We studied the cell type-specific pathways altered in the AD profiles identified from the Knight ADRC cohort by leveraging snRNA-seq from a subset of the Knight ADRC participants. We used the top 5% of genes that showed the most specific cell-type expression profile in neurons, astrocytes, oligodendrocytes, oligodendrocyte precursor cells (OPC), and microglia. Then, we used this information to determine if any AD molecular profiles showed a stronger signature of cell type-specific dysregulation. These analyses showed that Knight-C4 was enriched in genes overexpressed in astrocytes and endothelial cells (38% and 23% for all hits, respectively; Fig 7A, left panel and S13 Data). Whereas the first may indicate reactive astrogliosis, the latter may suggest endothelial dysfunction in AD development, providing more evidence of vascular involvement in this disease. Pathways analyses of astrocytic genes identified astrocyte-related pathways (Fig 7B) including axon guidance ($p = 2.8 \times 10^{-05}$) and hedgehog signaling pathway ($p = 1.6 \times 10^{-06}$). Previous work has shown the role of astrocytes in axon guidance during development and repair [97] and hedgehog signaling pathways involved in neuroinflammation, neuronal cell differentiation, and neuronal death [98]. Examples of genes involved in these pathways are *FYN*, *SMO*, *KIF7*, and *GLI2*. Many of these pathways were also enriched in overexpressed genes in endothelial cells (S17A Fig).

Knight-C4 has also enriched in genes down-regulated in microglial and neuronal genes (Fig 7A, right panel and S13 Data). Neuronal genes showed an association with synaptic vesicle cycle ($p = 1.0 \times 10^{-06}$) and retrograde endocannabinoid signaling ($p = 3.6 \times 10^{-03}$) [99–101], and gap-junction ($p = 2.2 \times 10^{-02}$) formation known to play an essential role in intercellular metabolic communication and transmission across electrical synapses [102] (Fig 7C and 7D). Two AD biomarker genes, *SNAP25* and *SYT1*, were included in these pathways. Microglial down-regulated genes implicate AD-related pathways including phagosome ($p = 1.0 \times 10^{-06}$), herpes simplex virus 1 infection ($p = 2.0 \times 10^{-04}$)—previously associated with AD risk [61–64], gap junction ($p = 2.2 \times 10^{-02}$), and AD ($p = 5.4 \times 10^{-08}$) (S17B Fig). Two interesting examples of genes represented in these pathways are *C3* and *CD74*, which are implicated in AD pathology. Further explorations showed that Knight-C4 is significantly enriched in down-regulated genes related to homeostatic microglia including C-X3-C Motif

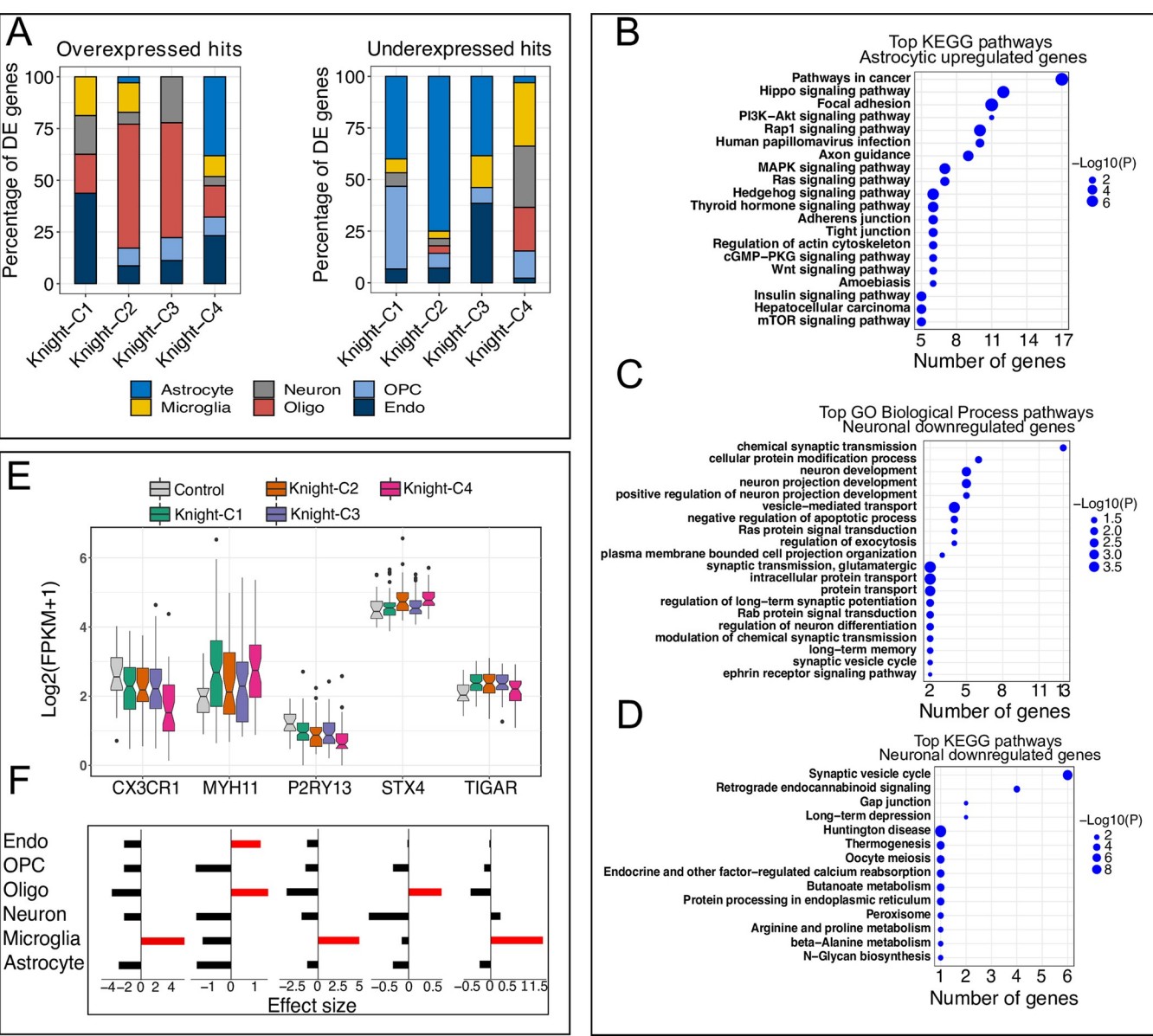

**Fig 7. Integrating single-nuclei data with cross-omics profiles identified cell type-specific genes and pathways.** (A) Bar plots showing the percentage of up- and down-regulated genes (clusters vs. control) identified in each cell type. (B) Top 20 KEGG pathways enriched in astrocytic up-regulated genes. (C) Top 20 GO biological process pathways enriched in neuronal down-regulated genes. (D) Top KEGG pathways enriched in neuronal down-regulated genes. (E) Boxplots show the transcriptomic profiles (in FPKM) of examples of cell type-specific genes identified in the Knight ADRC cohort. (F) The effect size of these examples was generated from single-nuclei data from the Knight ADRC participants. Red bars represent the cell type in which the gene is overexpressed. The data underlying panels A, E, and F can be found in **S1** Data. DE, differential expression; OPC, oligodendrocyte precursor cell.

Chemokine Receptor 1 (*CX3CR1*; $p = 5.10 \times 10^{-03}$, **Fig 7E** and **7F**) and purinergic receptor P2Y13 (*P2RY13*; $p = 1.2 \times 10^{-2}$, **Fig 7E** and **7F**).

Additionally, we identified genes dysregulated in specific clusters and cell types including the TP53-induced glycolysis regulatory phosphatase (*TIGAR*) overexpressed in microglia and up-regulated in Knight-C1 (**Fig 7E** and **7F**). *TIGAR* is a p53-inducible protein that regulates energy metabolism and oxidative stress. Among the up-regulated endothelial-specific genes in Knight-C1, we identified *MYOCD*, *CNN1*, *PLN*, *MYH11*, S*100A4*, *SLC13A4*, *DES*, implicating

pathways associated with cell cycle, junction, anion transport, Aβ clearance, and actin. In particular, *MYH11* overexpressed in Knight-C4 (**Fig 7E** and **7F**) has been reported to be associated with the risk of dementia [103]. Moreover, oligodendrocyte-specific genes dysregulated in Knight-C2 include *STX4* (Syntaxin-4), *ACTN4*, and *FBXO32* relating to pathways associated with vesicle docking, neural tube, junction, apoptosis, and actin. Of particular interest, *STX4* up-regulated in Knight-C2,4 (**Fig 7E** and **7F**) is critical for vesicle docking and the Wnt frizzled receptor (*FZD9*) and coreceptor (*LRP5*), regulating the synaptic vesicle cycle [31]. *STX4* has also been shown to be one of the disease risk genes in AD [104]. These results suggest that combining cross-omics data with single-cell resolution allows the uncovering of molecular profiles associated with AD features to unprecedented granularity and complexity.

## *SNCA* is a top hit in the multimodal cluster Knight-C4

Transcriptomic and proteomic assays showed *SNCA* among the top gene/proteins that best discriminate Knight-C4 versus Knight-C1-3 molecular profiles. *SNCA* transcriptomic levels were significantly down-regulated in Knight-C4 compared to Knight-C1-3 ($p = 2.0 \times 10^{-05}$) and control brains ($p = 2.2 \times 10^{-02}$; **Fig 8A**). Interestingly, we did not observe a significant association ($p > 0.05$) in unclustered AD compared to the control (**Fig 8A, left panel**). *SNCA* encodes alpha-synuclein (aSyn) protein, whose protein levels were marginally down-regulated in brains from Knight-C4 (**Fig 8B**) compared to controls ($p = 5.16 \times 10^{-02}$) and significantly lower in Knight-C2 compared to controls ($p = 7.8 \times 10^{-5}$) and Knight-C1,3,4 ($p = 6.2 \times 10^{-09}$).

 *SNCA*/*aSyn* is also down-regulated in MSBB-C1 (**Fig 8C** and **8D**) compared to the controls ($p = 1.8 \times 10^{-02}$ and $p = 1.0 \times 10^{-05}$ for transcriptomics and proteomics, respectively). Similarly, *SNCA* levels in MSBB-C1 were significantly lower than other AD ($p = 1.0 \times 10^{-02}$ and $p = 3.4 \times 10^{-03}$ for transcriptomics and proteomics, respectively, **Fig 8C** and **8D**). However, no significant differences in transcript or protein levels of *SNCA* were observed in MSBB-C2 compared to controls ($p > 0.05$, **Fig 8C** and **8D**). Down-regulation of *SNCA* in the MSBB-C1 donors was also seen in the frontal pole (BM10), superior temporal gyrus (BM22), and inferior frontal gyrus (BM44) brain regions (**S18A–S18C Fig**). Similarly, *SNCA* transcriptomic levels were significantly down-regulated in ROSMAP-C1 (**Fig 8E**) compared to the control ($p = 3.5 \times 10^{-02}$) and other AD cases ($p = 4.5 \times 10^{-03}$).

 *aSyn* is the major constituent of Lewy bodies, the histopathologic hallmark of PD and DLB. LB pathology is also present in most advanced AD cases including autosomal dominant AD and DS [105–111]. In addition, previous work has demonstrated the relationship between *aSyn* and AD pathologies including Aβ plaques and tau aggregations using different mouse models [112,113] and implicated *aSyn* pathology in neuronal/synaptic dysfunction and death [114,115]. Co-pathology with *aSyn* is present in 50% to 60% of AD brains [105,106,116]. However, it is not clear if it plays any role in AD pathophysiology. In animal models, it has been shown that adding *aSyn* to an amyloidosis model increased neuron loss that correlated with the progressive decline of cognitive and motor performance, and a "feed-forward" mechanism whereby Aβ plaques enhance endogenous α-syn seeding and spreading over time has been propose [112]. Mutations in *SNCA*, overexpression of wild-type *aSyn*, and injection of pre-formed *aSyn* fibrils (PFFs) are sufficient to cause neuropathology and parkinsonian syndromes in both humans and animal models [117–123].

 We also analyzed NanoString gene expression data (see Materials and methods) generated from a panel of 807 gene transcripts measured in brain tissue of A53T *aSyn* transgenic mice crossed onto *APOE* backgrounds [124]. The *SNCA* A53T mutation causes familial Parkinson's disease. Of the 807 genes included in the panel, 89 (11%) were classified as synaptic genes (SynGO consortium [47]). We performed DE analysis to compare mice with high pSyn

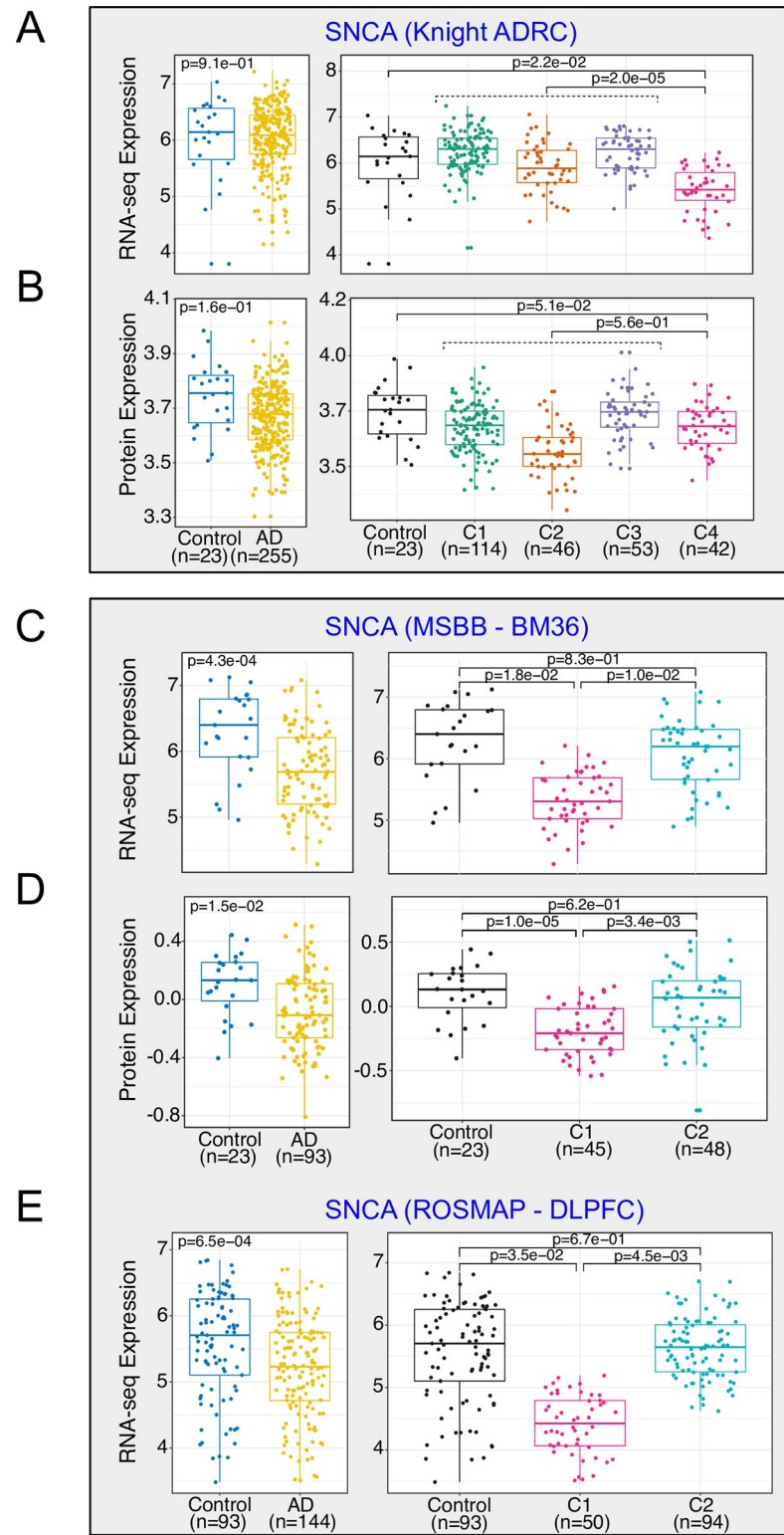

**Fig 8. Cross-omics integration identified alpha-synuclein levels down-regulated in AD participants with worse cognitive function common across multiple brain regions.** (A) Transcriptomic profiles (in FPKM) of *SNCA* across 4 clusters (right) and for all AD cases (left) in the Knight ADRC cohort. (B) Proteomic profiles of *SNCA* across 4 clusters (right) and all AD cases (left) in the Knight ADRC cohort. (C) Transcriptomic profiles of *SNCA* across 4 clusters (right) and all AD cases (left) in the MSBB BM36 cohort. (D) Proteomic (TMT) profiles of *SNCA* across 4 clusters

(right) and all AD cases (left) in the MSBB BM36 cohort. (E) Transcriptomic profiles of *SNCA* across 4 clusters (right) and all AD cases (left) in ROSMAP DLPFC cohort. The data underlying this figure can be found in S1 Data. AD, Alzheimer disease; DLPFC, dorsolateral prefrontal cortex; MSBB, Mount Sinai Brain Bank; ROSMAP, Religious Orders Study and Memory and Aging Project; TMT, tandem mass tag.

pathology (defined as >5% coverage by IHC) to mice with low pathology (<5% coverage by IHC, see Materials and methods). We identified 255 DEGs including 13 synaptic genes. We then interrogated which synaptic genes in DE in Knight-C4 were also dysregulated in the A53T aSyn-APOE transgenic mice (S19 Fig). We identified that all 13 synaptic genes identified as dysregulated in the mouse model were also dysregulated in Knight-C4 compared with other AD cases (Fisher $p = 3.6 \times 10^{-09}$) (S19A Fig). These results suggest that the profile of Knight-C4 is reproducible in mouse models of aggregation and neurodegeneration. Compared to the control, only 5 synaptic genes were shared between the 3 datasets (S19B Fig). These findings were limited because the nanoString panel contains only 12% of the significant genes in Knight-C4. These results show that integrating data from multimodal clusters in humans with molecular data from an established mouse model of neurodegeneration due to *aSyn* provides new clues about the role of known proteins in AD pathophysiology.

## Cross-omics integration identified CSF synaptic biomarkers for the molecular staging of AD

Knight-C4 exhibits significant dysregulation of synaptic genes and association with AD pathology but critically, is also associated with poor cognitive performance. Thus, we hypothesized that synaptic changes associated with AD molecular multimodal profiles could also reflect multiple stages of AD. To evaluate this, we selected those genes dysregulated in Knight-C4 (and ROSMAP-C1) and examined the effect size differences between the early-AD versus control compared to that of the late-AD versus control (Fig 9A) among the ROSMAP cohort. Interestingly, we observed a concordant change of synaptic gene dysregulation in early and late-AD compared to controls; however, the effect was more pronounced at later stages (Fig 9A and 9B, $R^2 = 0.96$, $p < 2.2 \times 10^{-16}$, slope = 0.75). This was consistent among 170 synaptic genes including *IGF1*, *NRXN3*, and *YWHAZ* (Fig 9C). Co-expression analyses (see Materials and methods) identified a core module (S20 Fig, **turquoise color**) highly enriched in interacting proteins including *SNAP25*, *APP*, and *SNCA* (S21A Fig). Subsequent pathway analyses identified several AD-related pathways including 160 protein–protein interaction hub proteins (S21B Fig and S14 Data). Additional gene interaction analyses showed that those proteins have a strong interaction including physical and genetic interactions (S21C–S21E Fig).

Next, we sought to determine whether proteins encoded by these genes could be potential CSF biomarkers for staging AD pathology and monitoring disease progression. To explore this hypothesis, we performed survival analyses including individuals with CDR change from 0 to 0.5 and using CSF proteomic (Somascan) data from 553 participants from the Knight ADRC. Our results showed that 11 of 13 (84%) proteins encoded by synaptic genes including *IGF1*, *NRXN3*, and *YWHAZ* (Fig 9D) were associated with an increased risk of dementia progression. Several studies have shown the involvement of *IGF1* in regulating signaling pathways altered in AD and neurodegeneration [125–129] and its potential as an early CSF/plasma biomarker of disease onset [130]. However, the association between *IGF1* and the increased risk of dementia progression in different stages of AD has not been studied. Likewise, the low expression of *NRXN3* is associated with an increased AD risk [131,132]. The *YWHAZ/ 14−3 −3* protein, present in CSF patients, has also been shown to be involved in the rapid progression of dementia [133] and identified as a promising therapeutic target for AD intervention

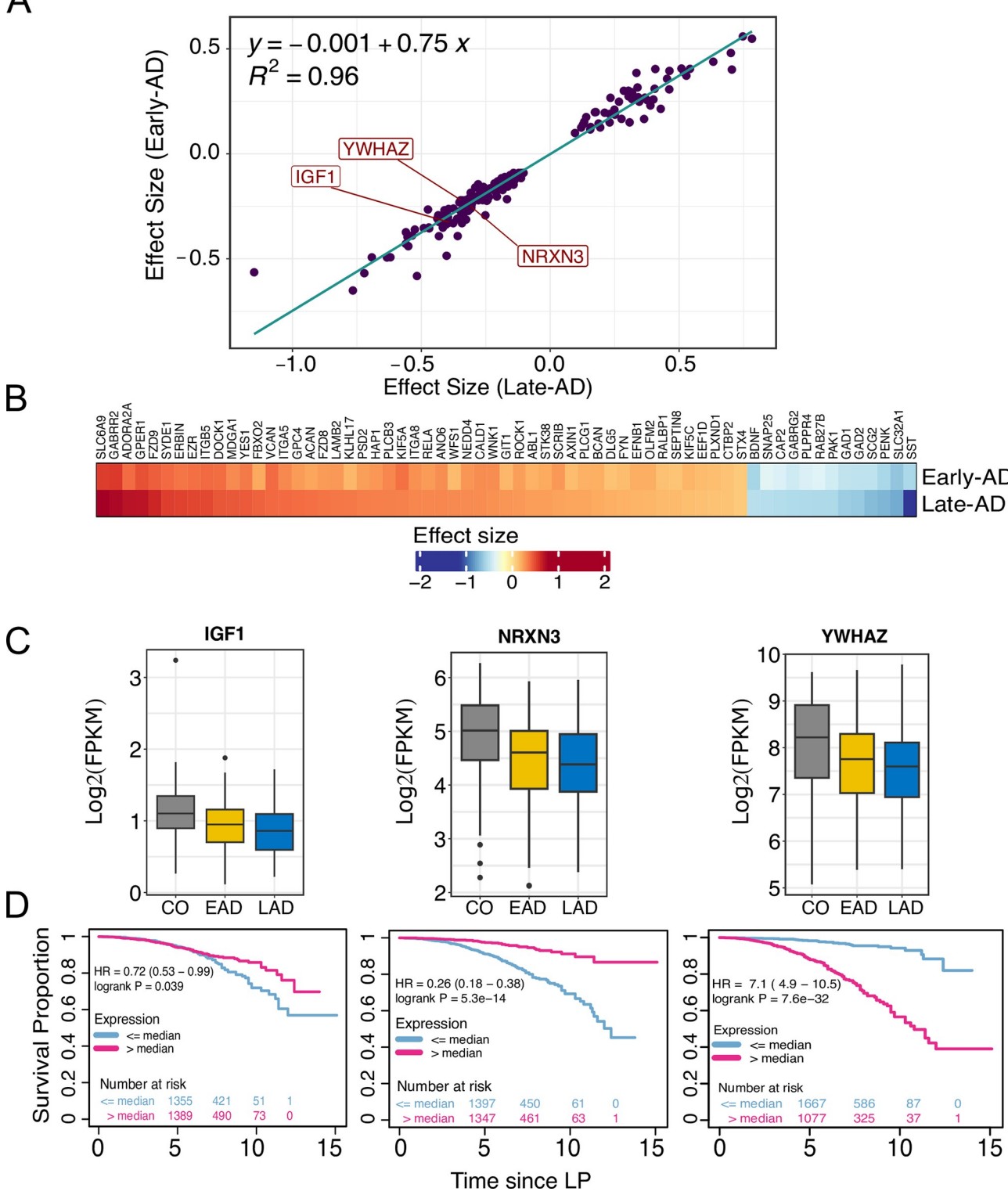

**Fig 9. Synaptic dysregulation at multiple stages of AD identified CSF synaptic biomarkers for the molecular staging of AD.** (A) Scatterplot showing the correlation of effect size between early-AD (EAD) and late-AD (AD) for common dysregulated synaptic genes in Knight-C4 and RSOSMAP. (B) Heatmap showing the effect size similarity of the most significant synaptic genes detected in early and late-AD. (C) Boxplots showing a consistent expression (transcriptomic) pattern between control, early-AD, and late-AD for genes *IGF1*, *NRXN3*, and *YWHAZ*. (D) Kaplan–Meier plots showing the association between *IGF1*, *NRXN3*, and *YWHAZ* genes and an increased risk of dementia progression using CSF proteomics data. The data underlying panel C can be found in S1 Data. AD, Alzheimer disease; CSF, cerebrospinal fluid.

[134]. These results show that the single molecules identified in our multimodal clusters from postmortem tissue could help identify candidate CSF biomarkers for monitoring AD progression in live patients. Furthermore, we investigated whether these 11 CSF proteins could also identify molecular differences among AD patients and their temporal progression patterns using the Subtype and Stage Inference (SuStaIn) machine learning technique [135,136] (see Materials and methods). Out of the 11 proteins, 8 were selected based on QC analyses to be used with SuStaIn. Two subtypes of AD that exhibited distinctive progression patterns based on multiple z-score thresholds (see Materials and methods) were identified (S22A and S22B Fig). While *IGF1* tends to become abnormal in late stages in Subtype1 (S22C Fig), it is predicted to be abnormal in early stages in Subtype2 (S22D Fig). These observations may suggest that *IGF1* might be a good candidate as an early CSF biomarker for AD, as it has been reported previously [130]. In addition, *NRXN3* (a synaptic cell–cell adhesion molecule) occurs at an early stage in Subtype2 (S22D Fig) but later in Subtype1 (S22C Fig) which may suggest an earlier increased AD risk for patients in Subtype2 than the ones in Subtype1. In contrast, *PLXNC1* started to become abnormal in the early stages in Subtype1 and then became stable before beginning to show abnormality again at later stages (S22C Fig), whereas in Subtype2 it is predicted to be abnormal at later stages (S22D Fig). *PLXNC1* was shown to be associated with neuronal loss [137] and therefore, these observations may indicate that *PLXNC1* might be a good candidate as a biomarker for monitoring the synapse and neuronal loss. Altogether, these results show that these CSF protein levels identified distinct temporal progression patterns in AD and detected molecular heterogeneity among AD cases and in the presence of control subjects with CDR = 0.

## Discussion

In the current study, we leveraged machine learning approaches to integrate high-throughput cross-omics data from controls and AD cases in multiple cohorts and brain regions. Our results highlight the utility of integrating cross-omics data and indicate that cross-omics signatures can better capture differences and discriminate molecular variations in complex and heterogeneous diseases such as AD. In the discovery stage, we used data from the parietal cortex, an understudied brain region affected in later stages of AD [16,41,42]. The parietal cortex was chosen to study dysregulated molecules in the presence of AD pathological lesions before a higher burden of Aβ plaques and NFTs occurs along extended periods of time, and possibly additional compensatory effects confound initial changes associated with etiology. In addition, the parietal cortex typically captures the more initial molecular changes in AD etiology compared to other severely affected regions (e.g., DLPFC) [16], making it a suitable region for our study.

We found a distinct molecular profile associated with worse cognitive function, earlier age at onset/death, and dysregulated neuronal/synaptic pathways in the parietal cortex. Our results suggest that this cross-omics molecular profiling captures novel molecular events not fully captured by β-amyloid or tau staging as previously reported [138,139]. Similar to previous studies [3], we did not find a significant association of *APOE* ε4 or PRS with any molecular profile, suggesting that these molecular profiles are not driven by genetic factors associated with AD risk. Through a rigorous and stringent replication using multiple cohorts (MSBB, ROSMAP), we also determined that this molecular profile replicates in additional cortical regions (parietal cortex, DLPFC, PHG) where it associates with tau pathology. Further, we demonstrated that this molecular profile is associated with poor clinical and cognitive function in these studies, offering new insights into the genes/proteins/metabolites associated with cognition decline and pathways associated with worse prognosis.

Notably, we observed that these donors with poor cognitive function, earlier onset, and younger age at death showed significantly increased cortical cellular proportions of astrocytes and reduced neurons. This subset of AD-afflicted participants is significantly enriched in dysregulated synapse-related genes and pathways, which may reflect greater synaptic losses including dopaminergic synapse and glutamatergic pathways. Although the involvement of dopamine in AD is still debatable, several studies have shown an association between AD pathologies (e.g., Aβ deposition) and dopaminergic dysfunction [140–148]. Similarly, increasing evidence has demonstrated an association between glutamate alterations and AD pathology [149–152]. In addition, transcriptomic and proteomic DE analyses identified several genes associated with AD including *SNAP25*, *GFAP*, and *CLU*, which are more noticeably dysregulated in the distinct molecular profiles. We showed that lower levels of CSF *CLU* are associated with an increased risk of dementia progression in an extended cohort from the Knight ADRC. Moreover, metabolic differential abundance analyses identified significant metabolites associated with biologically relevant AD pathways, most of which are related to major neurotransmitter pathways. These brains also exhibited a metabolic profile associated with disease duration, supporting its implication in the later stages of AD. These novel molecular findings can be leveraged to identify new biomarkers and potential therapeutic targets—not necessarily related to amyloid and tau—that might be effective even in the later stages of the disease.

By integrating snRNA-seq with the cross-omics profiles identified in brains, we extracted additional insights supporting cell-type distinct patterns implicated in molecular AD profiles. For example, we determined that *CX3CR1*, a homeostatic microglia gene, is significantly down-regulated in Knight-C4. The lower expression of *CX3CR1* may indicate fewer homeostatic microglia, leading to neuronal damage and loss. The association between homeostatic microglia function and the degree of neuronal cell loss is linked to AD [153]. More recently, the loss of *CX3CR1* was associated with microglial dysfunction illustrated by dampened TGFβ-signaling, increased oxidative stress responses, and dysregulated pro-inflammatory activation [154]. However, we could not replicate a significant overexpression of microglia-activated genes in additional cortical regions.

We identified *SNCA/a*Syn as one of the top molecules for discriminating AD profiles. We demonstrated that the transcriptomic/proteomic levels of *SNCA* were under-expressed in a subset of AD cases with more advanced tau pathology and worse cognition and replicated this finding in several cohorts and brain regions. Despite the substantial efforts in studying the association between this protein and AD pathology, the mechanisms by which *aSyn* contributes to the pathogenesis and progression of AD remain elusive. Several regulatory mechanisms that contribute to *aSyn* levels have been proposed. For example, previous studies based on primary neuronal cultures [155], mouse models [156], and live cell imaging [157] have shown that neuronal or synaptic activities regulate the aggregation of *aSyn*. The down-regulation of *SNCA/a*Syn in a subset of AD cases with unfavorable outcomes suggests that *SNCA* might be a marker of neuron/synapse losses that can be used for identifying AD stages.

We performed AD staging analyses to study the progression of AD and its effect on the synaptic activities using dysregulated synaptic genes in the Knight-C4 and ROSMAP-C1 profiles. Our data suggest that synaptic dysfunction occurs at the early stages of the disease, but it is more pronounced as AD progresses. Survival analyses from CSF proteomics from the Knight ADRC identified several synaptic genes associated with an increased risk of dementia, such as *IGF1*, *NRXN3*, and *YWHAZ*. Those proteins can be used as potential CSF biomarkers to monitor disease progression.

Our results are consistent with previous studies that report molecular correlates with AD clinical data and neuropathological staging [3,10]. We showed that AD heterogeneity cannot be entirely explained by neuropathological variables (e.g., β-amyloid or tau accumulation),

*APOE* ε4 allele carrier status, or even by differences in age or sex. Most of the previous studies have included a single-omic layer [2,3] capturing molecular changes in a single component of the biological cascade, or have analyzed a single AD cohort and a unique brain region (lacking independent replication) [10] as we believe that molecular changes and heterogeneity in AD are better understood when studied in multiple brain regions. We showed that distinct molecular profiles of AD are present in multiple brain regions by leveraging multiple AD cohorts (Knight ADRC, MSBB, ROSMAP) and multiple brain regions (parietal cortex, DLPFC, PHG), offering us substantial insights into AD (e.g., our findings of the significant enrichment of AD molecular profiles in synaptic genes and pathways that allow us to identify CSF synaptic biomarkers).

Although this study provides new insights for understanding the molecular heterogeneity in AD brains, our study has several limitations, highlighting critical future research directions. A limited number of brain metabolomic studies were available, and unfortunately were largely underpowered. In addition, the multiple omics data from different brain regions and cohorts were generated at different times on different samples using different platforms. Thus, future studies that consistently apply the same platforms on multiple brain regions and cohorts can provide novel and deeper findings. Large number of brain tissue, possibly from donors at multiple Braak staging, including better representation of those initial stages, coupled with quantitative neuropath will help elucidate further AD heterogeneity. We also had to integrate omics data using different platforms due to the need for more harmonization in the methods used by different groups. Independent replication is needed to validate and better understand the role of key drivers of these profiles using disease models (e.g., iPSC or mouse models) including the role of *SNCA* in AD and its association with worsening cognitive function in AD. Longitudinal data from CSF/plasma from large cohorts could provide novel insights into the role of dysregulated genes/proteins/metabolites in AD progression. Additional multi-omics studies exploring multiple brain regions, particularly resilient areas, would be instrumental in determining the temporal progression of protein changes throughout AD progression. We only used 1 tool (iClusterBayes) to integrate large multi-omics data; however, developing novel and more statistically robust tools is needed. We plan to continue integrating additional cohorts and omics data (e.g., epigenomics) as they become available. Another promising area for future exploration is integrating cross-omics data with drug repositioning datasets that may lead to identifying novel therapeutic potentials for AD, bringing AD closer to precision medicine.

In summary, we have shown the capability of cross-omics approaches and their superiority over single-omic analyses as a powerful resource that provides molecular insight into AD pathogenesis and identifies possible biomarkers for early synaptic dysfunction, cognition, and AD staging that may eventually enable precision medicine.

## Materials and methods

### Study cohorts

**Discovery (Knight ADRC).** Frozen postmortem parietal lobe tissue samples from the Knight Alzheimer Disease Research Center (Knight ADRC) participants were provided by the Knight ADRC Neuropathology Core. Written informed consent for research use was obtained from all participants or their family members. The informed consent was approved by the Institutional Review Boards of Washington University School of Medicine in St. Louis, and research was carried out in according to the approved protocols and to the principles of the Helsinki Declaration. Only sporadic AD participants and controls with transcriptomics, proteomics, and metabolomics data available were included in this study, resulting in 255 sporadic AD and 23 controls.

- RNA-seq: The data generation for the RNA-seq dataset has been previously described [11–13]. Briefly, total RNA was extracted from frozen parietal cortex tissue using a Tissue Lyser LT and purified using RNeasy Mini Kits (Qiagen). The Nanodrop 8000 (Thermo Scientific) and TapeStation 4200 (Agilent Technologies) were used to perform quality control of the RNA's concentration and purity, and degradation. The RNA integrity number (RIN) was calculated using an RNA 6000 Pico assay on a Bioanalyzer 2100 and TapeStation 4200 (Agilent Technologies). The software determines the RINe on the Bioanalyzer and TapeStation taking into account the entire electrophoretic trace of the RNA including the presence or absence of degradation products. The DV200 value is defined as the percentage of nucleotides greater than 200nt. All cDNA libraries were prepared using a TruSeq Stranded Total RNA Sample Prep with Ribo-Zero Gold kit (Illumina) and sequenced on an Illumina HiSeq 4000 using 2 × 151 paired-end reads at the McDonnell Genome Institute at Washington University in St. Louis. The average number of reads per sample was 43,195,080.

- Proteomics: Proteomic data from the Knight ADRC were generated on the SomaLogic platform (SomaLogic Operating Co., Boulder, CO) from frozen parietal cortical tissue samples (approximately 500 mg) and CSF samples. The SomaLogic platform is a multiplexed, aptamer-based protein quantification platform [158]. The platform measured 1,305 aptamers in total. The proteomic data generation details have been described previously [14].

- Metabolomics: Metabolomic data from the Knight ADRC were generated on the Metabolon Precision Metabolomics UPLC-MS/MS platform (Metabolon, Morrisville, United States of America) from 50 mg frozen parietal cortical tissue samples. The platform measured 880 metabolites from 9 "super pathways": amino acids, carbohydrates, cofactors and vitamins, energy, lipids, nucleotides, peptides, xenobiotics, and partially characterized molecules. Metabolon provided structural identity annotations for 815 metabolites; the remaining 65 were excluded from the final dataset. The metabolomics data generation details have been described previously [15].

- Single-nuclei (snRNA-seq): The data generation of snRNA-seq has been previously described in [16,159]. Briefly, 74 frozen parietal tissues were processed according to the "Nuclei extraction and library preparation" protocol described in [159]. The tissue was homogenized in this protocol, and the nuclei were isolated using a density gradient. The nuclei were then sequenced using the 10X Chromium single-cell Reagent Kit v3, with 10,000 cells per sample and 50,000 reads per cell for each of the 74 samples.

**Replication (MSBB, BM36).**

- RNA-seq: The RNA-seq raw data from The Mount Sinai Brain Bank (MSBB) study was downloaded from the Synapse portal (syn3157743). It is publicly available as part of the Accelerating Medicines Partnership for Alzheimer's Disease (AMP-AD). In brief, these data were generated and sequenced from RNA extracted from 4 postmortem brain regions including frontal pole (BM10), superior temporal gyrus (BM22), parahippocampal gyrus (BM36), and inferior frontal gyrus tissue (BM44) from 313 subjects. RNA-seq libraries were prepared using the TruSeq RNA Sample Preparation Kit v2 (Illumina, San Diego, California, USA). The rRNAs were depleted using the Ribo-Zero rRNA Removal Kit (human/mouse/rat) (Illumina). Single-end non-standard reads of 101 bp were generated by Illumina HiSeq 2500 (Illumina) [44]. The average number of reads per sample was 32,201,200. RNA-seq data from the region parahippocampal gyrus (BM36) were selected for the cross-omics integration approach because of the availability of proteomics data. Data from the remaining

regions were used to explore the identified profiles' expression profiles to determine whether these are consistent across cortical brain regions.

- Proteomics (TMT): The proteomics TMT data were obtained from the AMP-AD Synapse portal (syn25006650) processed by Johnson and colleagues [35]. The raw proteomic TMT data is publicly available at AMP-AD and can be downloaded from the Synapse portal (syn21347564). In short, before TMT labeling, the 198 samples were randomized by co-variates (protein quality, sample concentration, diagnosis, age, and sex) into 20 batches (10 cases per batch). The digested peptides were resuspended in 50 mM HEPES (pH 8.5) and labeled with the TMT 11-plex kit (Thermo Fisher) according to the manufacturer's protocol. In each batch, GIS samples were labeled using TMT channel 126. All 11 channels were mixed equally and desalted with a 100 mg C18 Sep-Pak column (Waters) for the subsequent fractionation [35,160].

  **Replication (ROSMAP, DLPFC).**

- RNA-seq: The RNA-seq raw data from The ROSMAP Study were downloaded from the Synapse portal (syn17008934) available from AMP-AD. Briefly, RNA-seq samples were extracted using Qiagen's miRNeasy mini kit (cat. no. 217004) and the RNase-free DNase Set (cat. no. 79254) and quantified by Nanodrop. Agilent Bioanalyzer evaluated quality. Library preparation was performed by poly-A selection followed by first strand-specific cDNA synthesis, then dUTP for second strand-specific cDNA synthesis, and fragmentation and Illumina adapter ligation for library construction. Illumina HiSeq generated paired-end read sequences with a length of 101 bp. Fifty-seven samples were submitted later to the platform and run on an updated protocol (batch 2 protocol). For consistency, we removed those samples from this cohort. The average number of reads per sample was 47,434,600. Only No Cognitive Impairment (NCI) and AD with NO other cause of cognitive impairment cases were used in this study. All cases with "unknown" BraakTau scores were discarded. Plaque density and Neurofibrillary tangle burden (PHF tau tangles) are a summary of AD pathology derived from 3 AD pathologies: neuritic plaques, diffuse plaques, and neurofibrillary tangles based on 5 regions: mid-frontal cortex, mid-temporal cortex, inferior parietal cortex, entorhinal cortex, and hippocampus. Tangles are identified by molecularly specific immunohistochemistry based on the mean of 8 regions: tangles_hip (hippocampus), tangles_ec (entorhinal cortex), tangles_mf (mid frontal cortex), tangles_it (inferior temporal), tangles_ag (angular gyrus), tangles_calc (calcarine cortex), tangles_cg (anterior cingulate cortex), angles_sf (superior frontal cortex).

- Metabolomics: Data from the ROSMAP were generated by the Duke Metabolomics and Proteomics Shared Resource, a member of the ADMC, using protocols published previously for blood samples [161–163]. A custom protocol developed for the brain samples can be found on Synapse at syn10235609. The DLPFC metabolomic data from the ROSMAP studies quantified on the Metabolon Precision Metabolomics platform and preprocessed by the ADMC as described in [164] were downloaded from Synapse in July 2021 (syn25878459). The platform measured a total of 1,055 metabolites in the ROSMAP dataset.

## Quantification and statistical analysis

RNA-seq QC, alignment, and gene expression quantification: All RNA-seq datasets from the discovery and replication were processed and aligned using our in-house RNA-seq pipeline (https://github.com/HarariLab/RNA-seq-Pipeline). Genome reference and gene models were

selected similar to the TOPmed pipeline (https://github.com/broadinstitute/gtex-pipeline/blob/master/TOPMed_RNAseq_pipeline.md). Reference genome GRCh38 and GENCODE 33 annotation, including the addition of ERCC spike-in annotations, were used. We excluded ALT, HLA, and Decoy contigs from the reference genome due to insufficient RNA-seq tools that properly handle these regions. Before alignment, the quality of raw read sequences for all libraries was assessed using FastQC (v0.11.9) [165]. All raw read sequences were aligned to the human reference genome (GRCh38) using STAR (v.2.7.1a) [166]. The alignment quality was evaluated using sequencing metrics such as reads distribution, ribosomal content, or alignment quality provided by STAR using Picard tools (v.2.8.2) [167]. All samples that failed to pass the QC or were outliers (Knight ADRC = 9, MSBB = 18, ROSMAP = 6) were removed from the downstream analyses. Raw read counts for transcripts and genes were generated using STAR and computed transcript/gene expression levels as normalized in FPKM (Fragments Per Kilobase of transcript per Million mapped reads) format.

Metabolomics QC and Quantification: Metabolomics data from Knight ADRC and ROSMAP were processed similarly. Full details of the QC process have been previously described [15]. Briefly, 188 metabolites with readings missing in over 20% of donors were excluded, readings were log10-transformed, means per metabolite were adjusted to zero, and outlier readings (outside 1.5xIQR) were excluded. Missing metabolite values in <20% of donors were replaced by their mean value. No samples were removed due to the missingness rate. Principal component analysis was performed with the FactoMineR R package [168] to identify outlier samples; 4 outlier samples were excluded. The final dataset consisted of 627 metabolites. A similar procedure was used to clean the ROSMAP metabolomic dataset. Metabolites without assigned structural identities and metabolites with greater than 20% missing readings were excluded, readings were log10-transformed, the mean of each metabolite's distribution was adjusted to zero, and outlier readings were removed. Two samples missing greater than 20% of readings were excluded. The final dataset consisted of 595 metabolites.

Proteomics QC and Quantification: Proteomic data for brain and CSF samples from the Knight ADRC were first filtered based on a limit of detection (LOD). Samples with greater than 15% outlier analytes (analyte readings below the LOD) were excluded. Analytes were then filtered based on scale factor difference, meaning the absolute value of the maximum difference between the calibration scale factor per aptamer and the median for each plate run. Analytes were excluded if their scale factor difference exceeded 0.5. Based on log10-transformation of protein readings, samples exceeding 1.5-fold of the interquartile range (IQR) per analyte were considered outliers. Analytes with greater than 15% outlier samples were excluded. Similarly, samples that were outliers for greater than 15% of analytes were excluded. Full details of the proteomic data QC have been described previously [14]. Missing analyte readings and removed outlier readings were imputed using the impute.knn function from the impute R package (v1.56.0) [169]. A plate-wise batch correction was performed using the ComBat function from the sva R package (v3.30.1) [170]. The final brain dataset comprised 1,092 analytes, and the final CSF dataset consisted of 713 analytes. Before using these data, analytes with missing expressions in >20% of donors were excluded, and missing expressions of those in <20% were replaced with their mean value.

The processed data for TMT proteomics from MSBB BM36 was obtained from AMP-AD (syn25006647) and processed by Johnson and colleagues [35]. In brief, 760 raw files generated from 19 TMT 11-plexes were analyzed using the Proteome Discoverer suite (version 2.3, Thermo Fisher Scientific). The UniProtKB human proteome database containing Swiss-Prot and TrEMBL human reference protein sequences was used to search mass spectra. Peptide spectral matches and peptides were filtered to a false discovery rate (FDR) of less than 1% using Percolator software. After spectral assignment, peptides were assembled into proteins

and filtered based on the combined probabilities of their constituent peptides to a final FDR of 1% [35]. Unique and razor (that is, parsimonious) peptides were considered for quantification. TMT reporter abundance was transformed into a ratio followed by $\log_2$ transformation. Quality control analysis was performed, and outliers were removed. Similar to the Knight ADRC, analytes with missing expressions in >20% of donors were excluded, and missing expressions of those in <20% were replaced with their mean value.

Single-nuclei (snRNA-seq) QC, alignment, and gene expression quantification: Full details of the data process of snRNA-seq data have been previously described in [16,159]. In short, CellRanger (v2.1.1 10XGenomics) [171] software was employed to align the sequences and quantify gene expression. Read sequences were aligned to a custom pre-mRNA reference constructed from genome assembly GRCh38 and generated as described by 10X Genomics technical manual. Filtering and QC analyses were performed individually using each subject's Seurat package (v2.20 and 2.30). All nuclei with high mitochondria gene expression were removed, and only genes expressed in >3 cells were kept for downstream analysis. Cells with <1,800 or >8,000 genes expressed in them were excluded. The data were normalized using the *LogNormalize* function that normalizes the gene expression measurements for each cell by the total expression, scales by a factor equal to the median counts of all genes, and log-transforms the expression.

## Clustering of AD participants

To cluster AD participants, iClusterBayes (Integrative clustering of multiple genomic data types) [43] available from the iClusterPlus R package (v1.22.0) [172] was employed. For each integration process, shared samples across omics datasets were extracted, and the top 2,000 most variant features were selected. iClusterBayes was performed by fitting multiple Bayesian models using different numbers of eigenfeatures (clusters) ranging from 1 to 10. We ran 22,000 Markov Chain Monte Carlo (MCMC) iterations for each, of which the first 12,000 were discarded as burn-in. The prior probability for the indicator variable gamma of each data set was set to 0.1, whereas the posterior probability cutoff was set at 0.5. The standard deviation of the random walk proposal for the latent variable was set to 0.5. The remaining parameters were left with the default values. We used the Bayesian information criteria (BIC) and deviance ratio to evaluate clustering solutions that maximize the compactness of the clusters (intra-cluster similarity).

## Cell proportion analysis

CellMix R package (v1.6.2) [173] employs multiple digital deconvolution methods and was used to estimate cellular population structure from gene expression data (TPM) generated by the Salmon quantification tool. The machine learning model and the gene panel (set of marker genes expressed highly in specific cell types) were previously described in [13]. Four cell types (astrocytes, microglia, neurons, and oligodendrocytes) were tested using 2 algorithms, meanProfile [173] and ssNMF (semi-supervised nonnegative matrix factorization) [174]. The significant difference in cell proportion between clusters was computed using a generalized linear model (GLM) test and corrected by sex and age of death (AOD).

## Differential expression analysis of AD clusters

RNA-seq: DE analyses were performed to compare participants in each cluster to the control participants and other AD clusters using the DESeq2 R package (v.1.22.2) [175]. Our models were adjusted for sex, age at death, and the percentage of astrocytes and neurons. Lowly expressed genes were removed to enhance our confidence in the differentially expressed genes

we discovered. Only genes with expression >0.5 CPM (count per million) in at least 25% of samples in either group being compared were retained for downstream analyses. The FDR was estimated using Benjamini–Hochberg (BH) [176]. All genes with FDR <0.05 were considered differentially expressed genes.

Proteomics: Differential expression analyses between clusters were carried out with linear models (R function glm) with age at death and sex as covariates. *P*-values were adjusted using Benjamini–Hochberg (BH) [176].

Metabolomics: differential abundance analyses were also carried out with linear models, with age at death, sex, and PMI as covariates. *P*-values were adjusted using Benjamini–Hochberg (BH) [176].

Single-nuclei: Differential expression analyses were performed to compare the overexpression of each gene in each cell type relative to other cell types using glmmTMB R Package (v1.1.3). This approach uses a template builder (glmmTMB) to fit generalized linear mixed models. Our model was corrected by sex covariate. To ensure cell type-specific overexpression, for each cell type, only top genes whose effect size is greater than 90% percentile cutoff were retained for downstream integration analyses.

## Survival analyses

Kaplan–Meier survival analyses and Cox proportional hazards models were used to determine the association of different clusters with survival outcomes and also to determine the association with the risk of dementia progression using survival (v3.2–3) [177] and surviplot (v1.1.1) [178] R packages.

## Significance of overlap

The significance of overlap for all analyses was computed using Fisher's exact test available from the R stats package (v4.3.0) [179].

## Pathway analysis

All pathway analyses for transcriptomics and proteomics were performed using the EnrichR R package (v3.0) [180] using genes/proteins that were significantly up- and down-regulated between clusters versus control and between distinct clusters. Two widely used databases, gene ontologies (GOs), related to molecular function, biological process, and cellular components, and KEGG databases, were used. All pathways with an adjusted *p*-value <0.05 were considered statistically significant. Similar pathway analyses were performed for single-nuclei (snRNA-seq) using EnrichR using significantly up- and down-regulated genes in specific cell types. For metabolomics, pathway analyses were performed using MetaboAnalyst (v5.0) software [181] using the metabolites that were increased and decreased between clusters and the control and across clusters.

## Replication analysis

Our replication strategy was designed to focus on the molecular profiles identified in the discovery analyses of the Knight ADRC cohort by extracting the significant genes and proteins identified in the Knight ADRC cohort. Using this set of features, iClusterBayes was run on both datasets (MSBB and ROSMAP) using the same parameters used with the discovery cohort.

### NanoString gene expression analysis

The animal work, data generation, and full details of the QC processes have been previously described [124]. Briefly, A53T αSyn-Tg mice were crossed with human *APOE2*, *APOE3*, or *APOE4* knockin mice or *Apoe* knockout mice to generate A53T mice homozygous for one of the 3 human *APOE* alleles or completely null for *Apoe*. Animal procedures were performed in accordance with protocols approved by the Institutional Animal Care and Use Committee at Washington University School of Medicine (Animal Welfare Assurance #D16-00245 and Protocol Number: 22–0098). Sample sizes and genotypes for NanoString gene expression analysis were selected based on αSyn pathology and biochemistry results. RNA was isolated from 12-month-old A53T/EKO ($n = 10$), A53T/E2 ($n = 2$), and A53T/E4 ($n = 10$) mouse midbrain. Seven hundred ninety-three transcripts were quantified with the NanoString nCounter multiplexed target platform using a customized chip based on the Mouse Neuroinflammation panel (www.nanostring.com). The geometric mean of negative control lanes was subtracted from gene transcript counts, and gene expression data were normalized to the geometric mean of positive control lanes and a list of housekeeping genes [124]. QC analyses, including principal components analyses, were performed using nSolver 4.0 and the Advanced Analysis 2.0 plugin (NanoString). For differential expression analysis, pSyn pathology was expressed as low (<5% coverage by IHC) or high (>5% coverage by IHC) for each animal, and APOE genotype and pSyn pathology were selected as predictor covariates. Differentially expressed analyses were performed using linear regression models in R with sex as covariates. *P*-values were adjusted using Benjamini–Hochberg (BH) [176].

### AD staging analysis

Clinical and pathological data including Reagan, Cerad, and Braak scores as well as final consensus cognitive diagnosis (cogdx) from the ROSMAP cohort, were used to classify AD cases into early-AD and late-AD as well as other clinical statuses such as presymptomatic, mild cognitive impairment (MCI), and control. Differential expression analyses comparing early and late AD cases to the control were performed using the DESeq2 R package (v.1.22.2) [175]. Dysregulated synaptic genes from the Knight ADRC profile (Knight-C4) overlapped with the dysregulated synaptic genes from each comparison from ROSMAP, and the difference in effect size of the 2 groups (early-AD versus Control and late-AD versus Control) was examined. For subtype and stage inference, pySuStaIn: A Python implementation of the Subtype and Stage Inference algorithm [135,136] was used with a maximum number of clusters = 3. For each biomarker, 3 z-score thresholds (1 is mild affect, 2 is moderate affect, and > = 3 is severe affect) were identified to measure the effect on each biomarker. The maximum z-score threshold reached at the end of the progression was set to 5. The number of MCMC iterations was set to 100,000, and the start point was set to 25.

### Co-expression and gene interaction analyses

Co-expression analyses were performed using dysregulated synaptic genes in Knight-C4 using the WGCNA R package (v1.72–1) [182]. Gene interaction analyses were performed using GeneMANIA webserver [183].

### ROSMAP (RNA-seq)

Study data were provided by the Rush Alzheimer's Disease Center, Rush University Medical Center, Chicago. Data collection was supported through funding by NIA grants P30AG10161 (ROS), R01AG15819 (ROSMAP; genomics and RNAseq), R01AG17917 (MAP),

R01AG30146, R01AG36042 (5hC methylation, ATACseq), RC2AG036547 (H3K9Ac), R01AG36836 (RNAseq), R01AG48015 (monocyte RNAseq) RF1AG57473 (single nucleus RNAseq), U01AG32984 (genomic and whole exome sequencing), U01AG46152 (ROSMAP AMP-AD, targeted proteomics), U01AG46161(TMT proteomics), U01AG61356 (whole genome sequencing, targeted proteomics, ROSMAP AMP-AD), the Illinois Department of Public Health (ROSMAP), and the Translational Genomics Research Institute (genomic). Additional phenotypic data can be requested at www.radc.rush.edu. Study data were provided through NIA grant 3R01AG046171-02S2 awarded to Rima Kaddurah-Daouk at Duke University, based on specimens provided by the Rush Alzheimer's Disease Center, Rush University Medical Center, Chicago, where data collection was supported through funding by NIA grants P30AG10161, R01AG15819, R01AG17917, R01AG30146, R01AG36836, U01AG32984, U01AG46152, the Illinois Department of Public Health, and the Translational Genomics Research Institute.

## Alzheimer's Disease Metabolomics Consortium (ADMC)

The results published here are in whole or partly based on data obtained from the AD Knowledge Portal (https://adknowledgeportal.org). Metabolomics data is provided by the Alzheimer's Disease Metabolomics Consortium (ADMC) and funded wholly or in part by the following grants and supplements: NIA R01AG046171, RF1AG051550, 3U01AG024904-09S4, RF1AG057452, R01AG059093, RF1AG058942, U01AG061359, U19AG063744, and FNIH: #DAOU16AMPA awarded to Dr. Kaddurah-Daouk at Duke University in partnership with a large number of academic institutions. As such, the investigators within the ADMC, not listed specifically in this publication's author's list, provided data along with its preprocessing and prepared it for analysis but did not participate in the analysis or writing of this manuscript. A complete listing of ADMC investigators can be found at: https://sites.duke.edu/adnimetab/team/.

## MSBB

The results published here are in whole or in part based on data obtained from the AD Knowledge Portal (https://adknowledgeportal.org/). These data were generated from postmortem brain tissue collected through the Mount Sinai VA Medical Center Brain Bank. Dr. Eric Schadt from Mount Sinai School of Medicine provided them. Dr. Levey provided proteomics data from Emory University based on postmortem brain tissue collected through the Mount Sinai VA Medical Center Brain Bank provided by Dr. Eric Schadt from Mount Sinai School of Medicine.

## Supporting information

**S1 Fig. The deviance ratio (top) and the Bayesian information criterion—BIC (bottom) at each K (number of eigen features).** The optimal solution is observed with K = 3 when the BIC is the minimum and the deviance ratio is the maximum dividing our samples into 4 clusters (k +1).
(TIF)

**S2 Fig. Knight-C4 is not associated with polygenic risk scores (PRS).** Boxplots showing the polygenic risk scores (PRS) across 4 clusters using 4 different *p*-value thresholds showing no significant association between Knight-C4 and PRS scores. The data underlying this figure can be found in S1 Data.
(TIF)

**S3 Fig. Knight-C4 is not associated with genetic or neuropathological effects.** (A) The distribution of the APOE ɛ4 allele across 4 clusters shows no association with Knight-C4. (B) The distribution of Braak scores for tau across 4 clusters shows no significant correlation between Knight-C4 and Braak staging for tau. However, Knight-C4 exhibited more stage VI cases exceeding the mean distribution of Braak stage VI (dashed line) across all clusters. (C) The distribution of Braak scores for amyloid-β across 4 clusters showing no association with Knight-C4. The data underlying this figure can be found in S1 Data.
(TIF)

**S4 Fig. Dysregulated genes are significantly common across cohorts from multiple brain regions.** (A) Venn diagrams depicting the overlap between significant genes in Knight-C4 and MSBB-C1 cohorts, showing a replication of the molecular signature associated with worse cognitive outcomes. (B) Same as "A" but for significant proteins. (C) Venn diagrams depicting the overlap between significant genes in Knight-C4 and ROSMAP-C1 cohorts, showing a replication of the molecular signature associated with worse cognitive outcomes.
(TIF)

**S5 Fig. Molecular profiles of MSBB-C1 are associated with high Braak scores for tau and replicated in additional brain regions from the MSBB study.** (A) Boxplots showing MSBB-C1 are associated with high BraakTau scores. (B) Boxplots show a clear pattern of an association with the early age of death for cases in MSBB-C1. (C) Heatmaps showing the replication of transcriptomic profiles of MSBB-C1 from the BM36 region in additional regions including frontal pole (BM10), superior temporal gyrus (BM22), and inferior frontal gyrus (BM44) brain regions from the MSBB study. The data underlying panels A and B can be found in S1 Data.
(TIF)

**S6 Fig. Molecular profiles of ROSMAP-C1 are associated with multiple neuropathological variables.** (A) Boxplots showing ROSMAP-C1 association with high tangle densities. (B) Boxplots showing ROSMAP-C1 association with neurofibrillary tangle burdens. (C) Boxplots showing higher plaque densities for AD cases in ROSMAP-C1. The data underlying this figure can be found in S1 Data.
(TIF)

**S7 Fig. Knight-C4 is associated with more significant hits across multi-omics datasets.** (A) Venn diagrams show the differentially expressed genes detected in each cluster (cluster vs. control). (B) Same as panel "A" but for proteomics. (C) Same as panel "A" but for metabolomics.
(TIF)

**S8 Fig. Significant enrichment of dysregulated synaptic genes in Knight-C4 is replicated in additional brain regions.** (A) Venn Diagrams showing the overlap between significant genes in MSBB-C1 and synaptic genes from Fei and colleagues [46] and SynGO [47] datasets. (B) Same as "A" but overlaps with significant genes in ROSMAP-C1.
(TIF)

**S9 Fig. Top significant pathways dysregulated in AD molecular profiles from multiple cortical brain regions and AD cohorts.** (A). Top common KEGG pathways dysregulated in Knight-C4 and MSBB-C1 (Cluster vs. other AD cases). (B) Top common KEGG pathways dysregulated in Knight-C4 and ROSMAP-C1 (Cluster vs. other AD cases). (C) Top common GO biological process pathways dysregulated in Knight-C4 and ROSMAP-C1 (Cluster vs. other AD cases). (D) Top 20 KEGG pathways enriched in Knight-C4 down-regulated genes

(Knight-C4 vs. control). (E) Top 20 GO biological process pathways enriched in Knight-C4 down-regulated genes (Knight-C4 vs. control).
(TIF)

**S10 Fig. Top significant pathways dysregulated in AD molecular profiles from multiple cortical brain regions and AD cohorts compared to control cases.** (A) Top common KEGG pathways dysregulated in Knight-C4 and MSBB-C1. (B) Top common KEGG pathways dysregulated in Knight-C4 and ROSMAP-C1. (C) Top common GO biological process pathways dysregulated in Knight-C4 and MSBB-C1. (D) Top common GO biological process pathways dysregulated in Knight-C4 and ROSMAP-C1. (E) Top 20 KEGG pathways associated with the down-regulation of proteins in Knight-C4. (F) Top 20 GO biological process pathways enriched in Knight-C4 down-regulated proteins.
(TIF)

**S11 Fig. The decreased expression of *SNAP25* in Knight-C4 is replicated in MSBB and ROSMAP cohorts.** (A) Boxplots showing transcriptomic profiles of *SNAP25* across the 2 clusters (right) and all ADs (left) in the MSBB (BM36) cohort. (B) Boxplots showing proteomic (TMT) profiles of *SNAP25* across the 2 clusters as mentioned in "A." (C) Boxplots showing transcriptomic profiles of *SNAP25* across the 2 clusters (right) and all ADs (left) in ROSMAP (DLPFC) cohort. The data underlying this figure can be found in S1 Data.
(TIF)

**S12 Fig. *GFAP* proteomic levels are dysregulated across distinct clusters.** *GFAP* protomeric levels are significantly increased in Knight-C2 compared to the control and other AD cases (right). The data underlying this figure can be found in S1 Data.
(TIF)

**S13 Fig. The increased level of *GFAP* is replicated in MSBB and ROSMAP datasets.** (A) Boxplots showing the transcriptomic profiles of *GFAP* across the 2 clusters (right) and all AD cases (left) in the MSBB (BM36) cohort. (B) Same as "A" but for the proteomic profiles (TMT). (C) Same as "A" but in ROSMAP (DLPFC) cohort. The data underlying this figure can be found in S1 Data.
(TIF)

**S14 Fig. Differential abundance and pathways analyses identified metabolites decreased in Knight-C4 and ROSMAP-C1.** (A) Boxplots showing the decrease of metabolomics profiles of gamma-aminobutyrate (GABA) in Knight-C4 (left) and ROSMAP-C1 (right). (B) Same as "A" but for choline metabolite. The data underlying this figure can be found in S1 Data.
(TIF)

**S15 Fig. Differential abundance and pathway analyses identified metabolites increased in Knight-C4 and ROSMAP-C1.** (A) Boxplots showing the increase of metabolomic profiles of tryptophan in Knight-C4 (left) and ROSMAP-C1 (right). (B) Same as "A" but for glucose metabolite. The data underlying this figure can be found in S1 Data.
(TIF)

**S16 Fig. Metabolic differential abundance and pathway analyses identified multiple dysregulated carbohydrate/sugar and sphingolipid metabolites.** (A) Boxplots showing the decrease of metabolomic profiles of citrate in Knight-C4 (left) and ROSMAP-C1. (B) Boxplots showing the decrease of metabolomic profiles of NAD+ in both Knight-C4 (left) and ROSMAP-C1 (right). (C) Boxplots showing the increased metabolism levels of sphingosine in both Knight-C4 (left) and ROSMAP-C1 (right). The data underlying this figure can be found in

S1 -Data.
(TIF)

**S17 Fig. Integrating single-nuclei data with cross-omics profiles identified cell type-specific pathways.** (A) Top 20 KEGG pathways enriched in endothelial up-regulated genes (Knight-C4 vs. Control). (B) Top 20 KEGG pathways enriched in microglia down-regulated genes (Knight-C4 vs. Control).
(TIF)

**S18 Fig. Down-regulation of *SNCA* is replicated in additional brain regions from the MSBB study.** (A) Boxplot showing the transcriptomic profiles of *SNCA* in MSBB-C1 from the frontal pole (BM10) brain region. (B) Same as "A" but from superior temporal gyrus (BM22). (C) Same as "A" and "B" but from the inferior frontal gyrus (BM44) brain region from MSBB. The data underlying this figure can be found in S1 Data.
(TIF)

**S19 Fig. The synaptic dysregulation in Knight-C4 is partially recapitulated in mouse models.** (A) Venn diagram showing the overlap between significant genes in Knight-C4 compared to other clusters, significant genes from NanoString gene expression data from A53T αSyn mouse models, and synaptic genes from SynGO dataset. (B) Same as "A" but using significant genes in Knight-C4 compared to the control.
(TIF)

**S20 Fig. Gene dendrogram and module colors generated by WGCNA showing 2 distinct gene networks (gray = 64, turquoise = 209).**
(TIF)

**S21 Fig. Co-expression and pathway analyses identified strong gene interaction patterns between dysregulated synaptic genes.** (A) The gene interaction network for the turquoise module shows high interactions between dysregulated synaptic genes such as *SNAP25*, *SNCA*, and *APP*. (B) Barplot showing the top 25 hub proteins extracted from protein–protein interaction (PPI) hub proteins pathways. (C) Gene interaction network showing the relatedness of the 25 dysregulated hub proteins. The plot demonstrates multiple strong interactions including physical and genetic interactions among the query genes (25 hub proteins) and other genes. (D) *YWHAB* sub-network shows the genes highly connected with YWHAB, some of which are examples of either synaptic or other AD-related genes. (E) Same as "D" but for *SNCA* sub-network.
(TIF)

**S22 Fig. Molecular differences among AD patients and their temporal progression patterns.** (A) Markov chain Monte Carlo (MCMC) trace generated by SuStaIn showing periodic patterns of the likelihood among MCMC samples in each subtype. (B) Histograms of the model likelihood generated by SuStaIn showing a model with a clear 2 subtypes. (C) Positional variance diagrams showing the progression patterns (SuStaIn stages) of Subtype1 for the 8 CSF biomarkers. The colors in each stage represent the effect on the CSF biomarker levels where white is unaffected; red is mildly affected (z-score of 1); magenta is moderately affected (z-score of 2); and blue is severely affected (z-score of 3 or more). (D) Same as "C" but for Subtype2.
(TIF)

**S1 Table. Demographic characteristics of AD and control cases.**
(XLSX)

**S1 Data. The individual numerical values underlying Figs** 2C, 2E, 2F, 3B, 3C, 3E, 3F, 4A–4C, 5B, 5C, 5E, 7A, 7E, 7F, 8A–8E **and** 9C **as well as** S2, S3A–S3C, S5A, S5B, S6A–S6C, S11A–S11C, S12, S13A–S13C, S14A, S14B, S15A, S15B, S16A–S16C, **and** S18A–S18C **Figs.**
(XLSX)

**S2 Data. Cell proportions for the Knight ADRC, MSBB (BM36), and ROSMAP (DLPFC) cohorts.**
(XLSX)

**S3 Data. Differentially expressed genes in the Knight ADRC cohort.**
(XLSX)

**S4 Data. Differentially expressed genes/proteins in the MSBB (BM36) cohort.**
(XLSX)

**S5 Data. Overlap of significant genes between discovery cohort (the Knight ADRC) and replication cohorts (MSBB and ROSMAP cohorts).**
(XLSX)

**S6 Data. Differentially abundant genes/metabolites in ROSMAP (DLPFC) cohort.**
(XLSX)

**S7 Data. Summary of the significant features detected between each cluster and the control.**
(XLSX)

**S8 Data. Differentially expressed proteins in the Knight ADRC cohort.**
(XLSX)

**S9 Data. Differentially abundant metabolites in the Knight ADRC cohort.**
(XLSX)

**S10 Data. Number of significant features between each cluster and the others.**
(XLSX)

**S11 Data. Overlap between significant genes in the Knight ADRC, MSBB (BM36), ROSMAP (DLPFC) cohorts, and synaptic genes.**
(XLSX)

**S12 Data. Overlap between differentially abundant metabolites in the Knight ADRC and ROSMAP cohorts.**
(XLSX)

**S13 Data. Number and percentage of significant features specific to each cell type across 4 Knight clusters.**
(XLSX)

**S14 Data. List of protein–protein interaction (PPI) hub proteins.**
(XLSX)

## Acknowledgments

We thank all the participants, their families, the many involved institutions, and their staff, whose help and participation made this work possible. This work was supported by access to equipment made possible by the Hope Center for Neurological Disorders and the Departments of Neurology and Psychiatry at Washington University School of Medicine.

## Author Contributions

**Conceptualization:** Richard J. Perrin, Bruno A. Benitez, Oscar Harari.

**Data curation:** Abdallah M. Eteleeb, Brenna C. Novotny, Christopher Sohn, Eliza Dhungel, Logan Brase, Fabiana Farias, Kristy Bergmann, Joseph Bradley, Joanne Norton, Jen Gentsch, Fengxian Wang.

**Formal analysis:** Abdallah M. Eteleeb, Brenna C. Novotny, Carolina Soriano Tarraga, Christopher Sohn, Eliza Dhungel, Aasritha Nallapu, Jared Buss.

**Funding acquisition:** John C. Morris, Bruno A. Benitez, Oscar Harari.

**Investigation:** Abdallah M. Eteleeb.

**Methodology:** Abdallah M. Eteleeb, John C. Morris, Celeste M. Karch, Richard J. Perrin, Bruno A. Benitez, Oscar Harari.

**Project administration:** Bruno A. Benitez, Oscar Harari.

**Resources:** Albert A. Davis, Oscar Harari.

**Software:** Abdallah M. Eteleeb.

**Supervision:** Bruno A. Benitez, Oscar Harari.

**Writing – original draft:** Abdallah M. Eteleeb.

**Writing – review & editing:** Abdallah M. Eteleeb, Albert A. Davis, John C. Morris, Celeste M. Karch, Richard J. Perrin, Bruno A. Benitez, Oscar Harari.

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
