## [Editor Report · Decision Letter 0]

20 Sep 2023

Dear Dr Harari, 

Thank you for submitting your manuscript entitled "Brain cross-omics integration in Alzheimer's disease" for consideration as a Research Article by PLOS Biology and apologies for the delay in getting back to you. Some of our academic editors were travelling so it took me a bit longer to get feedback from them on your submission.

Your manuscript has now been evaluated by the PLOS Biology editorial staff as well as by an academic editor with relevant expertise and I am writing to let you know that we would like to send your submission out for external peer review.

Once your full submission is complete, your paper will undergo a series of checks in preparation for peer review. After your manuscript has passed the checks it will be sent out for review. To provide the metadata for your submission, please Login to Editorial Manager (https://www.editorialmanager.com/pbiology) within two working days, i.e. by Sep 22 2023 11:59PM.

Kind regards,

Christian

Christian Schnell, PhD

Senior Editor

PLOS Biology

cschnell@plos.org

---

## [Decision Letter · Decision Letter 1]

14 Nov 2023

Dear Dr Harari,

Thank you for your patience while your manuscript "Brain cross-omics integration in Alzheimer's disease" was peer-reviewed at PLOS Biology. It has now been evaluated by the PLOS Biology editors, an Academic Editor with relevant expertise, and by several independent reviewers. 

In light of the reviews, which you will find at the end of this email, we would like to invite you to revise the work to thoroughly address the reviewers' reports.

As you will see below, the reviewers think that the topic of the study is important and overall agree that the study is well executed and provides important insights. However, in particular Reviewer 1 raises a number of major concerns which need to be addressed, including that the two axes of subtyping - disease stages and variants - should be considered.

Given the extent of revision needed, we cannot make a decision about publication until we have seen the revised manuscript and your response to the reviewers' comments. Your revised manuscript is likely to be sent for further evaluation by all or a subset of the reviewers.

**IMPORTANT - SUBMITTING YOUR REVISION**

*Re-submission Checklist*

*Published Peer Review*

*PLOS Data Policy*

*Blot and Gel Data Policy*

Sincerely,

Christian

Christian Schnell, PhD

Senior Editor

PLOS Biology

cschnell@plos.org

REVIEWS:

Reviewer's Responses to Questions

**Do you want your identity to be public for this peer review?**

Reviewer #1: No

Reviewer #2: Yes: Michael Hawrylycz

Reviewer #3: No

Reviewer #1: This is a very interesting study proposing a multi-omics Alzheimer's disease stratification that is cross-validated across independent datasets. Several analyses for clarifying the biological distinctiveness of the identified subtypes were performed, including comparisons with snRNA, cellular profiles, clinical data, etc.

While the manuscript presents an extensive amount of relevant analyses, it also has some major shortcomings that preclude its publication at the current state. Some of the limitations, as the first mentioned below, are also very well discussed in the "subtyping" or "stratification" field, with key recent methodological advances/solutions proposed that are unfortunately not considered here. Similarly, the literature review is quite restricted and lacks essential antecedent research.

Major comments:

- A critical point is that the subtyping was based on direct clustering on the omics data (the features), without controlling in any way for the participants' disease stages. As it has been shown (e.g., Young et al, 2018, Nat. Comms) two relevant "axes" of variability/heterogeneity coexist in every diseased population, one cause by the differences in diseases stages and other by disease subvariants. Trying to detect any of these "axes" independently without considering the other is a common error in many studies focused on progression and heterogeneity. The identified subtypes here are very likely contaminated with variability from having subjects at different stages of disease advance. It may be the case that they are primarily capturing subjects' proximity in the disease's long timeline, and not similarity in disease mechanisms or subvariants. The authors should consider this important point, i.e., how to disentangle between real underlying subtypes and data clusters potentially confounded by similarities in disease severity.

- Also notable, the study misses relevant references, in particular those that have already presented extended multi-omics integration in AD, proposing both staging and subtyping across multiple populations. The authors should accurately describe the field and compare main results with previous. See for instance:

 Yasser Iturria-Medina, Quadri Adewale, Ahmed F. Khan, Simon Ducharme, Pedro Rosa-Neto, Kieran O'Donnell, Vladislav A. Petyuk, Serge Gauthier, Philip L De Jager, John Breitner, David A. Bennett, 2022. Unified Epigenomic, Transcriptomic, Proteomic, and Metabolomic Taxonomy of Alzheimer's Disease Progression and Heterogeneity. Science Advances. Vol 8, Issue 46. DOI: 10.1126/sciadv.abo6764

Also, the manuscript should make clear from the introduction the added value in the context of the previous work (no, not all previous AD stratifications have been restricted to a single omic, as claimed in the Introduction). As a suggestion, the authors may accentuate their use of single-nucleus RNA for subtypes-subtypes comparisons. 

- Across the manuscript's text, it is unclear and confusing if the snRNA-seq data was used or not for the subtypes discovery, or only the bulk transcriptomics. Even in the abstract, it reads like if it was considered, which would add value to the analyses in terms of novelty, but on Figure 1 it appears on the other/right side, for post-clustering comparisons. I strongly encourage the authors to make this the clearest possible (it is not clear also in the methods). And, if not considered, to clean all possible insinuations that it was (for example, when claiming in the abstract "to integrate high-throughput bulk and single-nucleus transcriptomic, proteomic, …").

Reviewer #2: The study Brain cross-omics integration in Alzheimer's disease by Eteleeb and colleagues is an important study and approach in the analysis and integration of molecular and clinical data in Alzheimer's disease. This topic is now of increasing interest given the significance of the disease, powerful computational methods, and the many available existing and new data sources.

The introduction is well written and surveys the significance of single cell genomics and AD biomarkers including molecular and trajectory analysis. Cross-omics as proposed in this study approaches reveal how complex biomolecular profiles change in association with a disease and the relationships and correlations between distinct classes of biological molecules. The authors employ a Bayesian integrative clustering method to integrate three omics datasets then applying multiple association analyses and including 144 association with clinical and neuropathological attributes (e.g., sex, age of death, age at onset, post-mortem interval, CDR, Braak amyloid stage, Braak neurofibrillary tangle stage), survival, and differential expression analyses to characterize AD molecular profiles. 

The figures are in general well done and Figure 1 presents a clear outline of the approach and methodology. The authors discovered four unique multimodal molecular profiles, one showing signs of poor

cognitive function, a faster pace of disease progression, shorter survival with the disease, severe neurodegeneration and astrogliosis, and reduced levels of metabolomic profiles. This profile shows similar cellular and molecular profiles in multiple affected cortical regions associated with higher Braak tau scores and significant dysregulation of synapse-related genes and endocytosis, phagosome, signaling pathways altered in AD early and late stages. These are interesting associations illustrating the power of this type of analysis. While the analysis methods presented are largely standard, there are done carefully and statistically valid.

The capability of cross-omics approaches and its superiority over single-omic analyses as a powerful resource that provides molecular insight into the AD pathogenesis and identifies possible biomarkers for early synaptic dysfunction, cognition, and AD staging. There is opportunity to put forth the the methodology presented as a general approach for data harmonization and analysis in AD/ADRD. There is considerably increased interest in this with several ongoing projects pertaining to data sources such as ROSMAP and the Million Veterans Program from the VA. These and other data source may additionally benefit from the approach here.

The networks analysis and pathway analysis including gene ontology results of Figure 9 are I believe less convincing. Many of these association are non very specific with limited p-value significance, and large network presentation such as in Figure 9 are not particularly revealing. The authors may consider tightening the paper with somewhat more focus on key results and the generality of the approach. All in all this is a well done study with interesting findings and potentially a valuable contribution to the analysis and harmonization of AD/ADRD data.

Reviewer #3: The manuscript "Brain cross-omics integration in Alzheimer's disease" by Eteleeb et al leveraged a large number of omics measurements to identify AD subtypes, followed by their detailed molecular analysis. The topic of AD subtyping has been gaining momentum lately. Perhaps this is afforded because of the accumulation of the omics data. The presented study is on par published on the same topic and is worthy contribution to the overall forum discussing the subtypes of the AD.

There are number of issues that I'd like to see addressed prior the publication.

1. Please discuss the choice of the parietal cortex for studying the AD. What was the rationale for selecting this brain region? What are the pros and cons compare to a more conventional one - DLPFC?

2. The main finding of the manuscript are the 4 clusters or subtypes of AD. Since the code was not available, I could only review the analysis based on the description in the methods section. Use of BIC and deviance criteria for selection of the number of clusters is a good choice. However, one discrepancy needs a little bit more effort to explain. The supplementary figure 1 states that the optimum is at number of clusters 3 (if I am reading the X axis right). The legend to the same figure states that K=3, thus the samples are divided into 4 clusters. This is confusing. Please double check the number of clusters and clarify the rationale and the arithmetic in the figure and the legend.

3. The discussion section is completely missing other studies that investigated the number of subtypes of AD. Please discuss your results within the context of other findings.

4. Legend of Figure 1 (page 6, line 128) states "genes=60,754". Human genome is limited to 20K or so genes. It isn't clear what authors referring to. Are these collective measurements of the canonical mRNA transcripts across 3 cohorts or number of transcripts including splice isoforms? I'll leave this up to authors discretion if they should switch to the term mRNA transcript rather than gene. 

5. The term "cross-omics" triggers a minor question. Did the authors decide to use the term "cross-omic" to set themselves apart from the rest of the studies that use the term "multi-omic"? Moreover, once sentence in the discussion section uses both terms in one sentence (page 28, line 664). If there is no good rationale for using a distinct term, please consider switching to "multi-omic". Though I leave this issue to author's discretion too. 

6. Please state how exactly code will be shared. I really hope it won't be "available upon request". Though ideally, I'd prefer the code available at the time of the review.

---

## [Decision Letter · Decision Letter 2]

23 Feb 2024

Dear Dr Harari,

Thank you for your patience while we considered your revised manuscript "Brain cross-omics integration in Alzheimer's disease" for consideration as a Research Article at PLOS Biology. Your revised study has now been evaluated by the PLOS Biology editors, the Academic Editor [and the original reviewers - EDIT AS APPLICABLE]. 

In light of the reviews, which you will find at the end of this email, we are pleased to offer you the opportunity to address the remaining points from the reviewers in a revision that we anticipate should not take you very long. In particular, we ask you to (i) discuss Reviewer 1's concerns about the clustering carefully as a limitation, (ii) address the comment regarding the use of Sustain, and (iii) make the advantages of using data from multiple brain regions clearer. We will then assess your revised manuscript and your response to the reviewers' comments with our Academic Editor aiming to avoid further rounds of peer-review, although might need to consult with the reviewers, depending on the nature of the revisions.

In addition, we'd like you to address the following editorial requests:

* We would like to suggest a different title to improve readability/accuracy: "High-throughput multi-omic data reveal molecular heterogeneity in Alzheimer's disease subtypes"

* In the abstract, can you please specify that "cohorts" refers to groups of human patients? Otherwise it may not be clear that you include data from human patients as well.

* Please include the full name of the IACUC/ethics committee that reviewed and approved the animal care and use protocol/permit/project license. Please also include an approval number.

* Please include information about the form of consent (written/oral) given for research involving human participants. All research involving human participants must have been approved by the authors' Institutional Review Board (IRB) or an equivalent committee, and must have been conducted according to the principles expressed in the Declaration of Helsinki.

* DATA POLICY:

Regardless of the method selected, please ensure that you provide the individual numerical values that underlie the summary data displayed in the following figure panels as they are essential for readers to assess your analysis and to reproduce it: 2C, 2E, 2F, 3B, 3C, 3E, 3F, 4A, 4B, 4C, 5B, 5C, 5D, 5E, 7A, 7E, 7F, 8, and 9C (and similar panels in the supplementary information)

* The data do not seem to be publicly available. Can you please, in the Data Availability Statement, describe which restrictions apply and the procedures to obtain access to the datasets?

* Thank you for depositing your code on github. Could you please assign a DOI so that the repository is citable and versioned for your paper? Zenodo is one of the available tools for this.

**IMPORTANT - SUBMITTING YOUR REVISION**

*Resubmission Checklist*

*Published Peer Review*

*PLOS Data Policy*

*Blot and Gel Data Policy*

Sincerely,

Christian

Christian Schnell, PhD 

Senior Editor

PLOS Biology

cschnell@plos.org

REVIEWS:

Reviewer #1: I believe that while the study involve an impressive amount of data, it still suffers from major pitfalls, starting at the methodological level, and that many of the authors' replies don't address effectively the raised points:

- Independently of if detecting subtypes or profiles or variants, the use of clustering directly on data from an heterogenous population is fundamentally wrong. Although the participants were neuropathologically confirmed as AD, they surely were at different points on the disease's long timeline (and having different level of neuropathological and clinical severity, plus of course different levels of molecular alterations just due to progression differences), and just throwing the data into clustering would actually imply a lot of confounding effects in the discovery analysis.

- We will always find clusters, or profiles, but these can be deeply affected or influenced by how aligned where the subjects at the time of analysis. If not at same or close stage, the clusters although statistically 'valid', will be biologically meaningless.

- Once the clusters are detected from a mixed population (in terms of progression), all the subsequent analysis are affected by it, and unfortunately all discoveries will be strongly biased. 

- BTW, the need to use Sustain on a very significantly small subset of markers is unjustified and somehow disconnected from the rest.

- The use of data from multiple brain regions is claimed as a novelty of the manuscript, however, in my opinion the authors don't really show the advantages or disadvantages of it. It is explicitly said that it is better, but it is not studied, and not shown with clarity.

Again, clusters (or profiles) and subsequent significant differences in post-hoc analyses will always appear. My impression is that, sadly, the authors try to reply/convince by points (more data, more analyses, more figures, etc.), but there is still an underlying lack of understanding on the meaning of biological profiles (or clusters or subtypes or subgroups, name is irrelevant) to study disease heterogeneity.

Reviewer #2 (Michael Hawrylycz): The authors have made a large effort to meaningfully address my critiques and the critiques of other reviewers, I believe, and the paper is much improved.

Reviewer #3: The authors addressed all of my raised concerns and improved the manuscript.

---

## [Editor Report · Decision Letter 3]

28 Mar 2024

Dear Dr Harari,

Thank you for the submission of your revised Research Article "Brain high-throughput multi-omics data reveal molecular heterogeneity in Alzheimer's disease" for publication in PLOS Biology. On behalf of my colleagues and the Academic Editor, Yejin Kim, I am pleased to say that we can in principle accept your manuscript for publication, provided you address any remaining formatting and reporting issues. These will be detailed in an email you should receive within 2-3 business days from our colleagues in the journal operations team; no action is required from you until then. Please note that we will not be able to formally accept your manuscript and schedule it for publication until you have completed any requested changes.

As you address the formatting requests to come, please attend to these last editorial requests:

* Please include the full name of the IACUC/ethics committee that reviewed and approved the animal care and use protocol/permit/project license. Please also include an approval number.

* Please include a statement that your study has been conducted according to the principles expressed in the Declaration of Helsinki.

PRESS

Sincerely, 

Christian

Christian Schnell, PhD

Senior Editor

PLOS Biology

cschnell@plos.org